# SAPipe: Staleness-Aware Pipeline for Data-Parallel DNN Training

**Yangrui Chen**
The University of Hong Kong
yrchen@cs.hku.hk

**Cong Xie**
ByteDance
cong.xie@bytedance.com

**Meng Ma**
ByteDance
meng.ma@bytedance.com

**Juncheng Gu**
ByteDance
juncheng.gu@bytedance.com

**Yanghua Peng**
ByteDance
pengyanghua.yanghua@bytedance.com

**Haibin Lin**
ByteDance
haibin.lin@bytedance.com

**Chuan Wu**
The University of Hong Kong
cwu@cs.hku.hk

**Yibo Zhu**
ByteDance
zhuyibo@bytedance.com

## Abstract

Data parallelism across multiple machines is widely adopted for accelerating distributed deep learning, but it is hard to achieve linear speedup due to the heavy communication. In this paper, we propose SAPipe, a performant system that pushes the training speed of data parallelism to its fullest extent. By introducing partial staleness, the communication overlaps the computation with minimal staleness in SAPipe. To mitigate additional problems incurred by staleness, SAPipe adopts staleness compensation techniques including weight prediction and delay compensation with provably lower error bounds. Additionally, SAPipe presents an algorithm-system co-design with runtime optimization to minimize system overhead for the staleness training pipeline and staleness compensation. We have implemented SAPipe in the BytePS framework, compatible to both TensorFlow and PyTorch. Our experiments show that SAPipe achieves up to 157% speedups over BytePS (non-stale), and outperforms PipeSGD in accuracy by up to 13.7%.

## 1 Introduction

Deep Neural Networks (DNNs) have achieved ground-breaking performance on a wide range of domains, such as computer vision (CV) [10, 17] and natural language processing (NLP) [29, 7]. Meanwhile, the model sizes and data volumes have grown exponentially, making DNN training time-consuming and resource-intensive. The most common approach to accelerate DNN training is to use data parallelism, scaling DNN training across multiple devices. Despite the substantial speedup, distributed machine learning systems with data parallelism often cannot fully utilize the computation resources and achieve linear scaling (*i.e.*, GPU number times single-GPU training speed), due to non-negligible communication overhead [31, 2, 23, 13].

Many recent studies have been devoted to developing communication acceleration techniques. Some works reduce communication traffic using gradient compression [2] or mixed-precision training [21],

36th Conference on Neural Information Processing Systems (NeurIPS 2022).

while others schedule communication to overlap it with computation. For example, ByteScheduler [23] and PACE [3] propose preemptive communication scheduling to hide the communication overhead within forward computation time. These communication scheduling approaches reduce the communication overhead without affecting the convergence of training, but still cannot fully hide communication when the communication-to-computation ratio is high.

A new direction to accelerate distributed DNN training has been explored, which intentionally introduces staleness to the training pipeline in order to further increase the overlap between communication and computation. For example, PipeSGD [19] uses the gradients from the previous iteration in stochastic gradient descent (SGD), resulting in fixed 1-step staleness.

Though introducing fixed staleness may fully overlap communication with computation for some DNNs, the staleness may also incur severe problems to the convergence of training. We observe significant accuracy degradation or even training divergence with only 1-step staleness of gradients in our experiments. We identify this key limitation and propose 2 solutions: 1) partial staleness, which introduces staleness to a limited number of layers; 2) staleness compensation, which compensates the 1-step staleness by predicting the gradient to be produced in the next iteration, with optimized implementation to reduce the overhead of prediction.

We design a performant and $\underline{\text{S}}$taleness-$\underline{\text{A}}$ware communication $\underline{\text{Pipe}}$line (SAPipe) system for accelerating distributed DNN training, which reduces the communication overhead in distributed training by overlapping communication with computation, and approaches the linear scaling.

The main contributions of this paper are as follows:

- We propose a partial staleness algorithm, which finds the minimal number of layers to introduce staleness to, so as to keep the training pipeline running without stall (§ 3.1).
- We adopt multiple staleness compensation techniques, including delay compensation, weight prediction and their combinations (§ 3.2).
- We propose an algorithm-system co-design, kernel fusion, and other runtime optimizations in SAPipe, that are especially designed and implemented to minimize the system overhead of partial staleness and staleness compensation (§ 3.3).
- We provide theoretical guarantees to show that SAPipe achieves the same convergence rate as vanilla SGD, and conditionally better error bounds compared to PipeSGD (§ 4).
- We demonstrate that SAPipe outperforms existing frameworks. SAPipe achieves up to 157% speedups over BytePS (non-stale), and outperforms PipeSGD in accuracy by up to 13.7% (§ 5).

## 2    Background

**Preliminaries.**    In distributed deep learning, we solve the following optimization problem with $n$ workers: $\min_{x \in \mathbb{R}^d} F(x)$, where $F(x) = \frac{1}{n} \sum_{i \in [n]} F_i(x) = \frac{1}{n} \sum_{i \in [n]} \mathbb{E}_{z_i \sim \mathcal{D}_i} f(x; z_i), \forall i \in [n]$, is the objective function, $x \in \mathbb{R}^d$ is the set of model parameters ($d$ is the total number of model parameters), $z_i$ is a mini-batch of data randomly sampled from the local data $\mathcal{D}_i$ on device $i$, and $f(\cdot)$ is loss function. A typical DNN is composed of $m$ layers, which are concatenated into a flattened vector $x$ for simplicity.

**Distributed training with data parallelism.**    Data parallelism partitions the training data onto multiple devices, *i.e.*, workers. Each worker propagates its local data through the model and calculate the loss (*forward propagation*). It uses the loss value to compute the gradients of each parameter (*backward propagation*), and aggregates them from all workers, before updated to the global model. To facilitate distributed training, the parameter server [18] and all-reduce [26] are the two most popular communication architectures for gradient aggregation. The detailed process is shown in Algorithm 1, highlighted in blue.

**Staleness pipeline.**  To hide communication time, previous works (*e.g.*, PipeSGD [19]) introduce 1-step staleness to the training pipeline, where the forward computation can progress without waiting for the gradient synchronization. To be more specific, the gradient aggregation of each layer is executed once its backward computation is finished, followed by the optimizer update using the aggregated gradient from the previous iteration. Thus, the gradient aggregation overlaps with not only the backward computation and optimizer update of the current iteration, but also the forward computation of the next iteration. The gradient aggregation initiated in the current iteration will be

| **Algorithm 1** Distributed Training / Staleness Training Pipeline (PipeSGD) | **Algorithm 2** Staleness-Aware Pipeline with Delay Compensation (SAPipe-DC) |
|---|---|
| 1: Initialize $x_0$ | 1: Initialize $x_0$ |
| 2: **for all** iteration $t \in [T]$ **do** | 2: **for all** iteration $t \in [T]$ **do** |
| 3:      **for all** workers $i \in [n]$ in parallel **do** | 3:      **for all** workers $i \in [n]$ in parallel **do** |
| 4:          Compute $\nabla f(x_{t-1}; z_{i,t})$, $z_{i,t} \sim \mathcal{D}_i$ | 4:          Compute $\nabla f(x_{t-1}; z_{i,t})$, $z_{i,t} \sim \mathcal{D}_i$ |
| 5:          **if** $t = 1$ **then** | 5:          **if** $t > 1$ **then** |
| 6:            Same as $t > 1$ Pass | 6:            $g_{i,t} = \nabla f(x_{t-2}; z_{i,t-1})$ |
| 7:          **else** | 7:            $g_t = \frac{1}{n} \sum_{i \in [n]} g_{i,t}$ |
| 8:            $g_{i,t} \leftarrow \nabla f(x_{t-1}; z_{i,t})$ | 8:            $\Delta x_t = x_{t-1} - x_{t-2}$ |
| 9:            $g_{i,t} \leftarrow \nabla f(x_{t-2}; z_{i,t-1})$ | 9:            $g_t^{DC} \leftarrow DC_\lambda(g_t, \Delta x_t)$ |
| 10:          $g_t = \frac{1}{n} \sum_{i \in [n]} g_{i,t}$ | 10:            $\triangleright$ DC is a func. defined in Equ. (1) |
| 11:          $x_t \leftarrow optimizer(x_{t-1}, g_t, \eta_t)$ | 11:          $x_t \leftarrow optimizer(x_{t-1}, g_t^{DC}, \eta_t)$ |

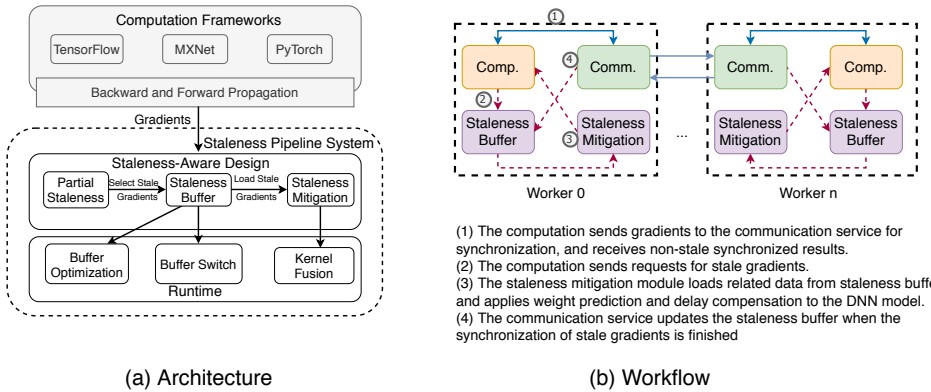

(a) Architecture          (b) Workflow

Figure 1: The architecture and workflow of SAPipe. Solid lines: data flow with all gradients. Dashed lines: data flow with stale gradients.

finished in the next iteration. We outline the conceptual distributed staleness training pipeline in Algorithm 1, highlighted in orange. Staleness pipeline results in delayed gradients, which may cause significant accuracy degradation of the converged model, as shown in our experiments (§ 5).

## 3 SAPipe Design

To address the convergence issue of staleness pipeline, we design a staleness-aware system, SAPipe, as shown in Figure 1(a), based on the following key designs.

**Partial Staleness.** Not all layers in a DNN model require a staleness training pipeline to fully hide communication within computation. Our partial staleness algorithm finds a minimal number of layers to be updated by stale gradients (§ 3.1), while updating the other layers without staleness.

**Staleness Compensation.** To further mitigate the problem caused by staleness of the layers chosen by the partial staleness algorithm, we compensate the staleness of the gradients by multiple approaches including delay compensation and weight prediction (§ 3.2).

**Optimized Runtime.** Both staleness training pipeline and staleness compensation incur additional overhead of computation and memory copy. To minimize such overhead, we adopt several system optimizations to achieve the high-performance runtime (§ 3.3).

Table 1: Notations

| Notation | Description | Notation | Description |
|---|---|---|---|
| $b_i$ | Duration of backward operator $i$ | $T, t$ | Total number and index of iterations |
| $u_i$ | Duration of forward operator $i$ | $g_t$ | Stochastic gradient $g_t = \frac{1}{n} \sum_{i \in [n]} g_{i,t}$ |
| $v_i$ | Duration of comm. operator $i$ | $(g_t)_j$ | The $j$th coordinate of $g_t$, $j \in [d]$ |
| $c$ | Completion time of operators | $(g_{i,t})_j$ | The $j$th coordinate of $g_{i,t}$, on worker $i$ |
| $n$ | Total number of workers | $(\nabla F_t)_j$ | The $j$th coordinate of $\nabla F(x_t)$, $j \in [d]$ |
| $m$ | Total number of layers | $\circ$ | Hadamard (coordinate-wise) product |
| $x$ | Model parameter $x \in \mathbb{R}^d$ | $d$ | The number of model parameters |

## 3.1 Partial Staleness

Due to the layer-wise structure of DNN models [1] and the reverse order of forward and backward passes, it only requires parts of the gradients to be stale to hide the communication overhead. Reducing the number of stale gradients mitigates the problems caused by staleness, improving convergence and keeping communication overhead hidden at the same time. We seek to find the minimal number of stale gradients such that there is no training pipeline stall, i.e., no waiting time for computing devices.

A naive way is to enumerate all possible combinations of gradients and check if they lead to training pipeline stall, which involves prohibitive complexity. Fortunately, the execution orders of the layers in sequential models are fixed, which provides the opportunity for efficient searching.

**Theorem 1.** *In a training pipeline without stall, if some forward layer is stale, then all its preceding forward layers are stale.*

Detailed proof of Theorem 1 is in the Appendix. Then we consider the case that the first $k$ layers need to be trained on staled gradients for fully overlapping its computation and communication, and formulate the following convex program for finding the minimal number of stale layers. Here the variables are defined in Table 1.

$$\text{minimize} \quad k$$

$$\text{subject to} \quad \sum_i v_i \leq \sum_i b_i + \sum_i u_i, \ \sum_{i=k+1}^{m} v_i \leq \sum_{i=1}^{m-1} b_i + \sum_{i=1}^{k} u_i, i = 1, \ldots, m;$$

$$0 \leq k \leq m, v_i > 0, b_i > 0, u_i > 0, \ i = 1, \ldots, m.$$

The first constraint aims to keep communication and computation fully overlapped. The second constraint ensures that the non-delayed gradients should be synchronized before the start of computation of the corresponding layers. We assume that the computation time of each layer remains the same under different solutions and the execution order follows FIFO. The above problem can be solved in $O(m^2)$ time by enumerating $k$. The optimal solution gives the minimal delayed gradients and ensures maximal training throughput.

In SAPipe, the first $k$ layers are updated with 1-step staleness, while the remaining layers are updated by vanilla distributed SGD without staleness (see Figure 2). The basic version of SAPipe (referred to as vanila SAPipe) is a mixture of distributed SGD and PipeSGD as given in Algorithm 1. For theoretical analysis, we add the following definition.

**Definition 1.** *Let $\kappa = \frac{d_{stale}}{d} \in [0, 1]$ denote the portion of model parameters updated by stale gradients, where $d_{stale}$ is the total number of model parameters contained in the $k$ layers that are updated with 1-step staleness.*

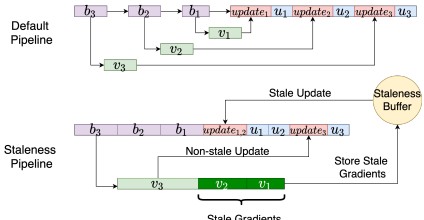

Figure 2: DNN training pipeline. The arrows denote dependencies between two operators.

---

[1] We focus on sequential models, and the discussion about extending our methods to DAG models can be found in the Appendix.

**Algorithm 3** Staleness-Aware Pipeline with Weight Prediction (SAPipe-WP)

1: Initialize $x_0$
2: **for all** iteration $t \in [T]$ **do**
3:     **for all** workers $i \in [n]$ in parallel **do**
4:         **if** $t = 1$ **then**
5:             Compute $\nabla f(x_0; z_{i,0})$, $z_{i,0} \sim \mathcal{D}_i$
6:         **else**
7:             $g_{i,t} \leftarrow \nabla f(\tilde{x}_{i,t-1}; z_{i,t-1})$                         ▷ Computed in the prev. iteration
8:             Option 1:  $\tilde{g}_{i,t} \leftarrow g_{i,t}$                               ▷ Local gradient
9:             Option 2:  $\tilde{g}_{i,t} \leftarrow g_{t-1}$                         ▷ Latest sync. gradient
10:            Option 3: $\tilde{g}_{i,t} \leftarrow DC(g_{t-1} - \frac{1}{n}g_{i,t-1}, \Delta x_t) + \frac{1}{n}g_{i,t}$, where $\Delta x_t = x_{t-1} - x_{t-2}$
11:            $\tilde{x}_{i,t} \leftarrow optimizer\,(x_{t-1}, \tilde{g}_{i,t}, \eta_t)$             ▷ 1-step-ahead weight prediction
12:            Compute $\nabla f(\tilde{x}_{i,t}; z_{i,t})$, $z_{i,t} \sim \mathcal{D}_i$ ▷ Synchronization is finished in the next iteration
13:            $g_t \leftarrow \frac{1}{n} \sum_{i \in [n]} g_{i,t}$
14:            $x_t \leftarrow optimizer\,(x_{t-1}, g_t, \eta_t)$

## 3.2 Staleness Compensation

We introduce two staleness compensation methods in our distributed training pipeline: delay compensation and weight prediction. Note that in SAPipe, they are only applied to the first $k$ layers of DNN, chosen by the partial staleness algorithm above, while the remaining layers are updated by normal distributed training procedure as shown in the blue part of Algorithm 1.

**Delay compensation (DC).** Inspired by DC-ASGD [33], we mitigate the effects of delayed gradients on the model with delay compensation. This method leverages Taylor expansion of the gradient function and efficient approximation of the Hessian matrix of the loss function. In the original design of DC-ASGD, the following delay-compensated gradient is used: $g_t \approx g_{t'} + \lambda g_{t'} \circ g_{t'} \circ (x_{t-1} - x_{t'-1})$, where $t > t'$, $g_t$ is the gradient evaluated on $x_{t-1}$, and $\lambda > 0$ is a hyperparameter. Note that the delay compensation above is a diagonal approximation of the full matrix form: $g_{t'} \circ g_{t'} \circ (x_{t-1} - x_{t'-1}) \approx g_{t'} g_{t'}^{\top} (x_{t-1} - x_{t'-1})$. In this paper, we prefer to use the full-matrix form of DC to avoid the additional error caused by the diagonal approximation:

$$g_t \approx DC_\lambda(g_{t'}, x_{t-1} - x_{t'-1}) = g_{t'} + \lambda g_{t'} g_{t'}^{\top} (x_{t-1} - x_{t'-1}). \tag{1}$$

The detailed algorithm of SAPipe with DC is shown in Algorithm 2. We will show that by compensating the 1-step staleness, SAPipe-DC achieves a lower error bound than vanilla SAPipe, and converges as fast as the non-stale baseline.

**Weight prediction (WP).** In a 1-step staleness pipeline, model weights are always updated using the synchronized gradients in the last iteration, resulting in convergence problems with inconsistent weight. The goal of WP is to estimate the 1-step-ahead weights for forward pass and backward pass, and to obtain 1-step-ahead gradients, in order to counteract the 1-step staleness. We provide three options for weight prediction: 1) WP with local gradient in the current step, 2) WP with the latest synchronized gradient, and 3) WP with all the above combined and DC.

The algorithm of SAPipe with WP is given in Algorithm 3. Line 5 computes the normal gradients in the first step. In the following steps, Line 7 retrieves the local gradient cached in each worker, and Line 8 to Line 10 provide the options for weight prediction. We apply an extra optimizer update using predicted gradients (Line 11) in each worker to predict 1-step-ahead weights, and conduct forward and backward propagation to obtain the 1-step-ahead gradients (Line 12). Line 13 finishes the synchronization of the gradients from the last step, and Line 14 uses them to update the model. Note that the delayed gradients in the last step are computed based on 1-step predicted weights. Therefore, the 1-step staleness is compensated.

## 3.3 Optimized Runtime

Figure 1(b) shows the workflow of the staleness pipeline system. SAPipe's staleness pipeline incurs extra overhead, namely: (1) the computation of staleness compensation; (2) data transfer between the staleness buffer and other modules. To minimize the system overhead, we present an algorithm-system co-design. We fuse the computation operations in DC and WP methods and other small weight update functions into several batched kernels. This saves a significant amount of kernel

launching time. We further optimize the usage of staleness buffer. We use a double buffer system [28] to handle stale gradients, reducing the transferring overhead, and minimize the memory usage by sharing the staleness buffer between computation and communication. We also switch the location of the staleness buffer between CPU memory and GPU memory according to the execution time of communication and computation pipeline stages. This overlaps the transferring overhead of the staleness buffer between communication and computation devices.

## 4 Theoretical Results

We establish theoretical guarantees of the convergence of SAPipe for smooth but non-convex problems, using SGD as the optimizer: $optimizer_{SGD}(x_{t-1}, g_t, \eta_t) = x_{t-1} - \eta_t g_t$, with a constant learning rate $\eta_t = \eta$.

### 4.1 Assumptions

**Assumption 1.** *(Smoothness) We assume that $f(x; z), \forall z$, are L-smooth: $\|\nabla f(x; z) - \nabla f(y; z)\| \leq L\|x - y\|, \forall x, y$, which implies $f(y; z) - f(x; z) \leq \langle \nabla f(x; z), y - x \rangle + \frac{L}{2}\|y - x\|^2$.*

**Assumption 2.** *For any stochastic gradient $g_{i,t} = \nabla f(x_{i,t-1}; z_{i,t}), z_{i,t} \sim \mathcal{D}_i$, where $\mathcal{D}_i$ is the local dataset on worker $i$, we assume bounded variance and $\ell_2$-norm: $\mathbb{E}[\|g_{i,t} - \nabla F_i(x_{i,t-1})\|^2] \leq V_1$, $\|g_{i,t}\|^2 \leq V_2, \forall i \in [n], t \in [T]$. Furthermore, gradients from different workers are independent of each other.*

**Assumption 3.** *Introduced in [30], we assume bounded gradient diversity: $\frac{\sum_{i \in [n]} \|\nabla F_i(x)\|^2}{\|\sum_{i \in [n]} \nabla F_i(x)\|^2} \leq \rho, \forall x$.*

The gradient diversity shows to what extent the local gradients on different workers are distinguished from each other. It is easy to check that Assumption 3 implies the bounded difference between the local and global gradients: $\frac{1}{n}\sum_{i \in [n]} \|\nabla F_i(x) - \nabla F(x)\|^2 \leq \left(\frac{\rho}{n} - 1\right) \|\nabla F(x)\|^2$.

**Assumption 4.** *For DC in Eqn. (1), we assume that the objective function $f(x)$ is twice-differentiable. Thus, we have the Taylor's approximation on $x' \in \mathbb{R}^d$ with bounded remainder: $\|\nabla f(x) - (\nabla f(x') + \nabla^2 f(x')(x - x'))\| \leq M\|x - x'\|^2$, where $\nabla^2 f(x')$ is the Hessian matrix evaluated on $x'$. We also assume that the Hessian approximation error is upper-bounded by $\Delta$, $\forall x \in \mathbb{R}^d$, i.e., $\|\nabla f(x)(\nabla f(x))^\top - \nabla^2 f(x)\|_2 \leq \Delta$, with vector-induced matrix norm $\|\cdot\|_2$.*

Note that the original paper of DC-ASGD proves that $\Delta \to 0$ when $t \to \infty$ under certain assumptions. Thus, it is reasonable to assume a bounded approximation error.

**Assumption 5.** *For simplicity, we assume that for partial staleness, the elements of every gradient are randomly and independently chosen to have 1-step staleness.*

**Assumption 6.** *There exists at least one global minimum $x_*$, where $F(x_*) \leq F(x), \forall x$. And we define the initial gap as $R_0 = F(x_0) - F(x_*)$.*

### 4.2 Convergence Analysis

We derive the following error bounds on the convergence of SAPipe under the above assumptions. All proofs can be found in Appendix A.

**Theorem 2.** *Under Assumptions 1, 2, 5 and 6, taking $\eta \leq \frac{1}{L}$, after $T$ iterations, for vanilla SAPipe without DC or WP, we have the following error bound: $\frac{1}{T}\sum_{t=1}^{T} \mathbb{E}\left[\|\nabla F(x_{t-1})\|^2\right] \leq \frac{2R_0}{\eta T} + \kappa Err_0 + Var_0$, where $Err_0 = L^2\eta^2 V_2$ and $Var_0 = \frac{L\eta V_1}{n}$.*

**Remark 1.** *Theorem 2 shows that the overall convergence error bound of vanilla SAPipe includes **2 main components: the gradient estimation error $Err_0$ and the variance error $Var_0$.** $Err_0$ is rooted in the 1-step staleness. $Var_0$ is incurred by the random sampling in stochastic gradients.*

**Remark 2.** *When no layer is updated with staleness, SAPipe is reduced to vanilla distributed SGD. In this case, the gradient estimation error $\kappa Err_0$ vanishes with $\kappa = 0$, which results in the exact error bound of vanilla distributed SGD. On the contrary, when all the layers are updated with staleness, SAPipe is reduced to PipeSGD with the gradient estimation error $Err_0$ and $\kappa = 1$.*

**Theorem 3.** *Under Assumptions 1, 2, 4, 5 and 6, taking a small enough $\lambda$ so that $\lambda \eta V_2 \leq 1 - \frac{1}{\sqrt{2}}$, and $\eta \leq \frac{1}{L}$, after $T$ iterations, for Algorithm 2 (SAPipe-DC), we have the following error bound:*

$\frac{1}{T} \sum_{t=1}^{T} \mathbb{E} \left[ \|\nabla F(x_{t-1})\|^2 \right] \leq \frac{2R_0}{\eta T} + \kappa Err_{DC} + (1 + \kappa)Var_0$, where $Err_{DC} = 128\eta^4 M^2 V_2^2 + 32\eta^2(1-\lambda)^2 L^2 V_2 + 32\eta^2 \lambda^2 \Delta^2 V_2 + 16\eta^3 \lambda^2 L V_2^3$.

**Remark 3.** *Regardless of the variance error, the main difference between the error bounds of vanilla SAPipe and SAPipe-DC is the gradient estimation error. Note that if we have a small enough Hessian estimation error yielding very small $M$ and $\Delta$, by taking small enough $\eta$ and making $\lambda \to 1$, the overall gradient estimation error of SAPipe-DC is smaller than that of PipeSGD, i.e., $Err_{DC} \leq Err_0$ under certain conditions. In other words, **SAPipe-DC has better convergence compared to PipeSGD when the Hessian approximation error is small enough**.*

**Theorem 4.** *Under Assumptions 1, 2, 3, 5 and 6, taking $\eta \leq \frac{1}{L}$, after $T$ iterations, for Algorithm 3 (SAPipe-WP) with Option 1, we have the following error bound: $\frac{1}{T} \sum_{t=1}^{T} \mathbb{E} \left[ \|\nabla F(x_{t-1})\|^2 \right] \leq \frac{2R_0}{\eta T} + \kappa Err_{WP-1} + Var_0$, where $Err_{WP-1} = L^2\eta^2 \left[ \left( \frac{\rho}{n} - 1 \right) V_2 + \left( 3 - \frac{\rho}{n} \right) V_1 \right]$.*

**Remark 4.** *The main difference between Theorem 2 and Theorem 4 is the gradient estimation error, i.e., $V_2$ versus $\left( \frac{\rho}{n} - 1 \right) V_2 + \left( 3 - \frac{\rho}{n} \right) V_1$. When $V_1$ and $\rho$ are small enough (i.e., $V_1 \leq \frac{2-\rho/n}{3-\rho/n} V_2$), SAPipe-WP with Option 1 produces a smaller error compared to the vanilla SAPipe, i.e., $Err_{WP-1} \leq Err_0$ under certain conditions. In other words, **SAPipe-WP with Option 1 has better convergence compared to vanilla SAPipe when the difference between the local datasets on different workers is small enough**.*

**Theorem 5.** *Under Assumptions 1, 2, 5 and 6, taking $\eta \leq \frac{1}{L}$, after $T$ iterations, for Algorithm 3 (SAPipe-WP) with Option 2, we have the following error bound: $\frac{1}{T} \sum_{t=1}^{T} \mathbb{E} \left[ \|\nabla F(x_{t-1})\|^2 \right] \leq \frac{2R_0}{\eta T} + \kappa Err_{WP-2} + Var_0$, where $Err_{WP-2} = \frac{L^2\eta^2}{1-2L^2\eta^2}(2V_1 + 2L^2\eta^2 V_2)$.*

**Remark 5.** *The main difference between Theorem 2 and Theorem 5 is the gradient estimation error, i.e., $V_2$ versus $\frac{1}{1-2L^2\eta^2}(2V_1+2L^2\eta^2 V_2)$. When $L$ is small enough (i.e., $L^2 \leq \frac{1-2V_1/V_2}{4\eta^2}$), SAPipe-WP with Option 2 produces a smaller error compared to vanilla SAPipe, i.e., $Err_{WP-2} \leq Err_0$ under certain conditions. In other words, **SAPipe-WP with Option 2 has better convergence compared to vanilla SAPipe when the objective function is "smooth" enough**.*

**Theorem 6.** *Under Assumptions 1, 2, 4, 5 and 6, taking $\eta \leq \frac{1}{L}$, after $T$ iterations, for Algorithm 3 (SAPipe-WP) with Option 3, we have the following error bound: $\frac{1}{T} \sum_{t=1}^{T} \mathbb{E} \left[ \|\nabla F(x_{t-1})\|^2 \right] \leq \frac{2R_0}{\eta T} + \kappa Err_{WP-3} + Var_0$, where $Err_{WP-3} = \eta^2 L^2[8\frac{V_1}{n} + 4\eta^4 M^2 V_2^2 + 2\eta^2 V_2(L^2(1-\lambda)^2 + \lambda^2 \Delta^2)]$.*

**Remark 6.** *The main difference between Theorem 2 and Theorem 6 is the gradient estimation error, i.e., $V_2$ versus $[8\frac{V_1}{n} + 4\eta^4 M^2 V_2^2 + 2\eta^2 V_2(L^2(1-\lambda)^2 + \lambda^2 \Delta^2)]$. When $\eta$, $M$, $\Delta$ are small enough and $n$ is large enough, **SAPipe-WP with Option 3 produces a smaller error compared to vanilla SAPipe, i.e., $Err_{WP-3} \leq Err_0$ under certain conditions**. Furthermore, $\lambda$ provides a trade-off between $L^2$ and $\Delta^2$. If the objective function is relatively "smooth" yielding $L < \Delta$, then $\lambda \to 0$ is preferred, and otherwise $\lambda \to 1$ is preferred. In practice, since both $L$ and $\Delta$ are unknown and depend on the model architecture and datasets, tuning $\lambda \in [0, 1]$ is required for better performance.*

**Corollary 1.** *For vanilla SAPipe, SAPipe-DC and SAPipe-WP with all 3 options, taking $\eta \propto \min \left( \frac{1}{\sqrt{T}}, \frac{1}{L} \right)$, we have $\frac{1}{T} \sum_{t=1}^{T} \mathbb{E} \left[ \|\nabla F(x_{t-1})\|^2 \right] \leq \mathcal{O} \left( \frac{1}{\sqrt{T}} \right) + \mathcal{O} \left( \frac{V_1}{n\sqrt{T}} \right)$, same as vanilla SGD.*

**Remark 7.** *The corollary above shows that vanilla SAPipe, SAPipe-DC and SAPipe-WP with all 3 options converge to a critical point where $\|\nabla F(x_{t-1})\| \to 0$, when $T \to +\infty$, and the error decreases when there are more workers. In general, vanilla SAPipe, SAPipe-DC and SAPipe-WP have the same overall convergence rate $\mathcal{O} \left( \frac{1}{\sqrt{T}} \right)$. However, the detailed gradient estimation error varies for different algorithms, as shown in Theorems 2, 3, 4, 5, and 6. Under certain conditions such as low gradient variance, low gradient diversity, good smoothness, low Hessian approximation error, and specific choices of $\eta$ and $\lambda$, one of these algorithms achieves the lowest convergence error. Hence, **hyperparameter tuning and algorithm selection are required in practice**.*

Table 2: Comparing model performance of SAPipe with baselines. We use perplexity (lower is better) as the metric for GPT-2, and accuracy (higher is better) for other models.

| Model | VGG16 | | ResNet50 | | GPT-2 | Transformer |
|---|---|---|---|---|---|---|
| Dataset | CIFAR-10 | ImageNet | CIFAR-10 | ImageNet | WikiText-2 | Multi30K |
| BytePS | 0.925 ± 0.0020 | 0.731 ± 0.0020 | 0.932 ± 0.0022 | 0.762 ± 0.0022 | 20.10 ± 0.23 | 0.673 ± 0.0033 |
| PipeSGD | 0.906 ± 0.0064 | 0.726 ± 0.0019 | 0.893 ± 0.0032 | 0.753 ± 0.0015 | 22.36 ± 0.33 | 0.531 ± 0.0044 |
| SAPipe-DC | 0.908 ± 0.0018 | 0.734 ± 0.0011 | 0.894 ± 0.0019 | **0.758 ± 0.0150** | 22.50 ± 0.48 | 0.526 ± 0.0031 |
| SAPipe-WP-OPT1 | **0.926 ± 0.0017** | 0.718 ± 0.0020 | **0.932 ± 0.0020** | 0.757 ± 0.0022 | 21.80 ± 0.25 | 0.663 ± 0.0059 |
| SAPipe-WP-OPT2 | 0.902 ± 0.0026 | 0.729 ± 0.0059 | 0.908 ± 0.0019 | 0.751 ± 0.0061 | 22.60 ± 0.09 | 0.662 ± 0.0007 |
| SAPipe-WP-OPT3 | 0.914 ± 0.0020 | **0.735 ± 0.0038** | 0.897 ± 0.0034 | **0.758 ± 0.0013** | **20.23 ± 0.32** | **0.668 ± 0.0032** |
| Over PipeSGD | 2% | 0.9% | 3.9% | 0.5% | 2.13 | 13.7% |

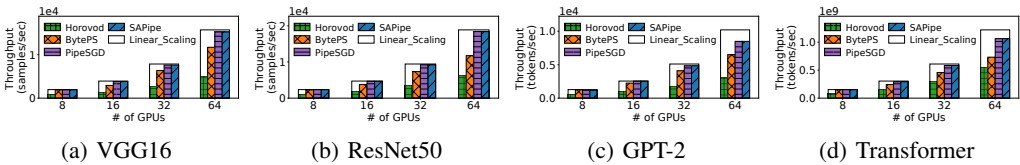

(a) VGG16  (b) ResNet50  (c) GPT-2  (d) Transformer

Figure 3: Training throughput.

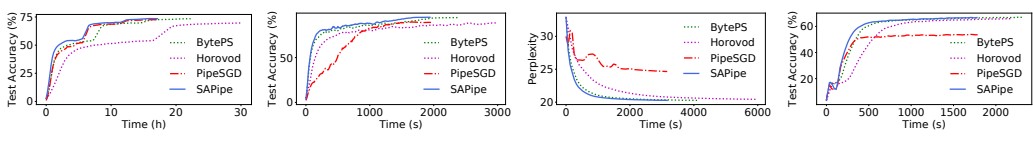

(a) VGG16, ImageNet  (b) ResNet50, CIFAR-10  (c) GPT-2, WikiText-2  (d) Transformer, Multi30K

Figure 4: Convergence of different models. x-axis is wall-clock training time, and y-axis is perplexity (lower is better) for GPT-2, and test accuracy (higher is better) for others.

# 5 Evaluation

## 5.1 Set-up

**Testbed.** We evaluate SAPipe [2] on 8 physical machines, each equipped with 90 CPU cores, 320GB memory, 8 Tesla V100 GPUs with NVLinks, and 100Gbps bandwidth between any two machines.

**Models and datasets.** We choose two CV models, VGG16 [27], ResNet50 [10], and two NLP models, pretrained GPT-2 [25], Transformer [29], as our benchmark models. The batch sizes per GPU are 128 images, 128 images, 80 tokens and 3200 tokens, respectively. We adopt SGD optimizer with 0.9 Polyak's momentum [24] and 5e-5 weight decay when training VGG16 and ResNet50 models, and Adam [14] optimizer with (0.9, 0.98) betas for NLP models. The global learning rates for VGG16, ResNet50 and GPT-2 are 0.1, 0.1, and 5e-5, respectively, and we follow the learning rate setting in [29] when training Transformer. SAPipe uses Option 3 in Algorithm 3 as the default staleness compensation method, with $\lambda$ empirically set as 0.2.

We train CV models on two datasets: (i) CIFAR-10 [16] and (ii) ImageNet [17]. We fine-tune the pretrained GPT-2 model on (iii) WikiText-2 language modeling dataset [20]. The Transformer model is trained on (iv) Multi30K [8] for WMT16 English-to-German Multimodal Translation task.

**Baselines.** We compare SAPipe with three state-of-the-art communication frameworks: (1) Horovod [26], a high-performance all-reduce paradigm; (2) BytePS [13], an optimized parameter-server architecture; (3) PipeSGD [19], a pipelined training framework with 1-step staleness. [3]. All baselines and SAPipe are run on PyTorch computation framework. [4]

---

[2] Code: https://github.com/ChenAris/sapipe.git

[3] Since PipeSGD is not open-sourced, we implement its staleness pipeline based on BytePS architecture.

[4] Magnified figures and additional experiment results are in the Appendix.

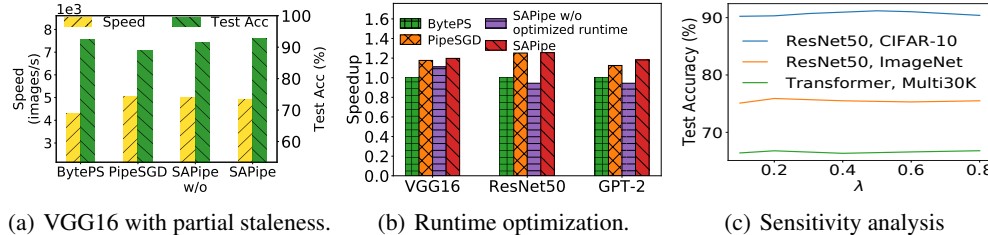

(a) VGG16 with partial staleness.  (b) Runtime optimization.  (c) Sensitivity analysis

Figure 5: Deep dive in SAPipe. "SAPipe w/o" denotes SAPipe without partial staleness.

## 5.2 Results and Analysis

**Convergence.**  Figure 4 shows the convergence curves when training DNN models with 16 GPUs. We observe that SAPipe converges much faster than baselines. It not only achieves the highest training throughput, but also incurs very little influence on model convergence. For example, SAPipe is roughly 35%, 67% and 21% faster for VGG16 than BytePS, Horovod and PipeSGD, respectively. Though PipeSGD has comparable training speed, it converges much slower across models, and decreases final accuracy by 13% with Transformer. This reveals the obvious error incurred by 1-step staleness.

**Staleness mitigation.**  Table 2 lists the final model performance of different frameworks. [5] SAPipe achieves comparable test accuracy and perplexity with non-stale baseline, BytePS. PipeSGD incurs significant performance drop on four models. With staleness compensation, SAPipe solves the convergence problem with staleness pipeline, while greatly boosting the training speed. Overall, SAPipe achieves significant improvement in accuracy/perplexity over PipeSGD of 2.6%, 3.5%, 13.7% and 2.13, for VGG16, ResNet50, Transformer and GPT-2, respectively.

We also compare different staleness compensation options of SAPipe with baselines. The best option varies among DNN training jobs. For example, "SAPipe-WP-OPT1" (*i.e.*, predicting weights using local gradients) achieves the highest test accuracy when training CV models on CIFAR-10, and "SAPipe-WP-OPT3" (*i.e.*, combined mitigation with WP and DC) is the best option for ImageNet, WikiText-2 and Multi30K datasets. Other options improve the convergence to some extent as compared to PipeSGD, but are suboptimal in these cases. The method to find the best staleness mitigation option for a given DNN task is our future work.

**Scalability.**  Figure 3 shows the throughput of baselines and SAPipe when training with different numbers of workers. SAPipe achieves up to 57% and 203% speedups over BytePS and Horovod in all settings. Horovod does not apply communication scheduling, which leads to the worst throughput. BytePS enables preemptive communication scheduling, but it does not overlap gradient synchronization of the first few layers with computation. Thus, the communication overhead in BytePS is still large for GPT-2 model. PipeSGD performs the best among baselines, while SAPipe still achieves up to 6% speedup over PipeSGD, thanks to our high-performance runtime optimization.

**Ablation study.**  We evaluate the effectiveness of each component of SAPipe. Figure 5(a) shows the throughput speedup and staleness mitigation with partial staleness when training VGG16 on CIFAR-10 dataset with 8 GPUs. We observe that with our partial staleness, SAPipe improves the final accuracy without slowing down training, by involving less delayed gradients.

We also compare SAPipe with BytePS, PipeSGD and SAPipe without runtime optimization when training DNN models with 32 GPUs, as shown in Figure 5(c). We use the speedup over the BytePS baseline as the throughput metric. "SAPipe w/o optimized execution" refers to a naive implementation of staleness-aware pipeline, which incurs non-negligible overhead for staleness mitigation. SAPipe achieves the highest throughput across all models, and the overhead of staleness mitigation decreases the training speed by 7% to 24% over SAPipe. Without runtime optimization, the staleness pipeline with mitigation methods could be slower than BytePS baseline by up to 6%.

**Sensitivity.**  Figure 5(d) varies the value of hyperparameter $\lambda$ in "SAPipe-WP-OPT3" method when training ResNet50 and Transformer. We observe that the staleness compensation method is not

---

[5]Horovod does not impact the convergence and has the same final accuracy as BytePS, so we omit it.

sensitive to the hyperparameter, and achieves the highest accuracy when it reaches 0.5, 0.2 and 0.2 when training on CIFAR-10, ImageNet and Multi30K datasets, respectively.

## 6 Related Work

**Communication optimization.** Many popular ML frameworks, such as TensorFlow [1] and PyTorch [22], enable overlapping communication with backward propagation by default. Recent works [12, 9, 23, 3] further overlap gradient synchronization with forward computation via tensor partitioning, at the cost of extra overhead. ASP [6] advocates asynchronous training to improve communication efficiency; without controlling the staleness, it leads to unstable convergence. SSP [11] is another asynchronous training protocol with bounded staleness. PipeSGD [19] restricts the staleness to one step in training. However, these methods do not mitigate the extra error caused by staleness, which degrades the model performance. We focus on synchronous pipelined training, without gradient compression. We minimize the number of stale layers, and propose multiple staleness compensation methods, achieving both high training throughput and comparable accuracy.

**Staleness mitigation.** Staleness mitigation is important for asynchronous SGD. Widely used approaches include staleness-aware rescaling of learning rate [32] and gradients [4], and delay compensation [33]. [15, 5] use linear weight prediction to narrow down the difference between models used in forward and backward passes for model parallelism. In our experiments, we find that simply using delay compensation or linear weight prediction in PipeSGD results in poor convergence, which calls for better strategies. Furthermore, the additional overhead incurred by staleness mitigation motivates our co-design of training algorithms and system optimization. However, the best staleness mitigation method varies in different DNN tasks, and this is the limitation of SAPipe that requires empirical experiments to find the optimal solution.

## 7 Conclusion

We present SAPipe, a performant and staleness-aware pipelined system to accelerate distributed DNN training without model performance loss. To fully overlap gradient synchronization communication with computation with minimal staleness, we introduce partial staleness, which restricts the number of layers learned with stale gradients. To further mitigate convergence issues caused by staleness, SAPipe adopts weight prediction and delay compensation. With an algorithm-system co-design, SAPipe achieves both better error bounds in theory, and high-performance runtime in practice.

## Acknowledgments and Disclosure of Funding

We are thankful to the anonymous NeurIPS reviewers for their constructive feedback. This work was supported in part by Hong Kong Innovation and Technology Commission's Innovation and Technology Fund (Partnership Research Programme with ByteDance Limited, Award No. PRP/082/20FX), and grants from Hong Kong RGC under the contracts HKU 17204619, 17208920 and 17207621.

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
