# Appendix

## A   Proofs

In this section we provide the detailed proofs of the theoretical analysis.

**Lemma 1.** *(General error bound) Under Assumption 1, taking $\eta \leq \frac{1}{L}$, in the $t^{th}$ step, for the distributed SGD with the general gradient estimator $g_{i,t}$ (the gradient estimator produced by the $i$th worker in the $t^{th}$ step), we have the following error bound in expectation conditional on $x_{t'}$ where $t' < t$:*

$$\mathbb{E}[F(x_t)]$$
$$\leq F(x_{t-1}) - \frac{\eta}{2}\|\nabla F(x_{t-1})\|^2$$
$$+ \frac{\eta}{2}\left\|\nabla F(x_{t-1}) - \frac{1}{n}\sum_{i\in[n]}\mathbb{E}[g_{i,t}]\right\|^2 + \frac{L\eta^2}{2}\mathbb{E}\left[\left\|\frac{1}{n}\sum_{i\in[n]}g_{i,t} - \frac{1}{n}\sum_{i\in[n]}\mathbb{E}[g_{i,t}]\right\|^2\right].$$

The theorem states that in each iteration, the loss function $F(x)$ is expected to decrease by $\frac{\eta}{2}\|\nabla F(x_{t-1})\|^2$, with the error caused by the variance of the stochastic gradient estimator $\frac{L\eta^2}{2}\mathbb{E}\left[\left\|\frac{1}{n}\sum_{i\in[n]}g_{i,t} - \frac{1}{n}\sum_{i\in[n]}\mathbb{E}[g_{i,t}]\right\|^2\right]$ (i.e., variance) and the error caused by the difference between the true gradient and the gradient estimator $\frac{\eta}{2}\left\|\nabla F(x_{t-1}) - \frac{1}{n}\sum_{i\in[n]}\mathbb{E}[g_{i,t}]\right\|^2$ (i.e., bias).

*Proof.* In the $t^{th}$ step, using $L$-smoothness, we have

$$F(x_t)$$
$$\leq F(x_{t-1}) + \langle \nabla F(x_{t-1}), x_t - x_{t-1}\rangle + \frac{L}{2}\|x_t - x_{t-1}\|^2$$
$$= F(x_{t-1}) + \left\langle \nabla F(x_{t-1}), -\eta\frac{1}{n}\sum_{i\in[n]}g_{i,t}\right\rangle + \frac{L}{2}\left\|-\eta\frac{1}{n}\sum_{i\in[n]}g_{i,t}\right\|^2.$$

Taking expectation on both sides conditional on $x_{t'}$ where $t' < t$, we have

$$\mathbb{E}[F(x_t)] \leq F(x_{t-1}) - \eta\underbrace{\left\langle \nabla F(x_{t-1}), \frac{1}{n}\sum_{i\in[n]}\mathbb{E}[g_{i,t}]\right\rangle}_{①} + \frac{L\eta^2}{2}\underbrace{\mathbb{E}\left[\left\|\frac{1}{n}\sum_{i\in[n]}g_{i,t}\right\|^2\right]}_{②}.$$

We bound the terms step by step.

First, for ①, we have

$$①$$
$$= \left\langle \nabla F(x_{t-1}), \frac{1}{n}\sum_{i\in[n]}\mathbb{E}[g_{i,t}]\right\rangle$$
$$= -\frac{1}{2}\|\nabla F(x_{t-1}) - \frac{1}{n}\sum_{i\in[n]}\mathbb{E}[g_{i,t}]\|^2 + \frac{1}{2}\|\nabla F(x_{t-1})\|^2 + \frac{1}{2}\left\|\frac{1}{n}\sum_{i\in[n]}\mathbb{E}[g_{i,t}]\right\|^2.$$

For ②, we have

$$②$$

$$= \mathbb{E}\left[\left\|\frac{1}{n}\sum_{i\in[n]}g_{i,t}\right\|^2\right]$$

$$= \mathbb{E}\left[\left\|\frac{1}{n}\sum_{i\in[n]}g_{i,t} - \frac{1}{n}\sum_{i\in[n]}\mathbb{E}[g_{i,t}] + \frac{1}{n}\sum_{i\in[n]}\mathbb{E}[g_{i,t}]\right\|^2\right]$$

$$= \mathbb{E}\left[\left\|\frac{1}{n}\sum_{i\in[n]}g_{i,t} - \frac{1}{n}\sum_{i\in[n]}\mathbb{E}[g_{i,t}]\right\|^2\right] + \left\|\frac{1}{n}\sum_{i\in[n]}\mathbb{E}[g_{i,t}]\right\|^2.$$

Combining the ingredients above together, we have
$\mathbb{E}[F(x_t)]$

$$\leq F(x_{t-1}) - \eta \underbrace{\left\langle \nabla F(x_{t-1}), \frac{1}{n}\sum_{i\in[n]}\mathbb{E}[g_{i,t}]\right\rangle}_{\text{①}} + \frac{L\eta^2}{2}\underbrace{\mathbb{E}\left[\left\|\frac{1}{n}\sum_{i\in[n]}g_{i,t}\right\|^2\right]}_{\text{②}}$$

$$\leq F(x_{t-1}) - \eta\left[-\frac{1}{2}\|\nabla F(x_{t-1}) - \frac{1}{n}\sum_{i\in[n]}\mathbb{E}[g_{i,t}]\|^2 + \frac{1}{2}\|\nabla F(x_{t-1})\|^2 + \frac{1}{2}\left\|\frac{1}{n}\sum_{i\in[n]}\mathbb{E}[g_{i,t}]\right\|^2\right]$$

$$+ \frac{L\eta^2}{2}\left[\mathbb{E}\left[\left\|\frac{1}{n}\sum_{i\in[n]}g_{i,t} - \frac{1}{n}\sum_{i\in[n]}\mathbb{E}[g_{i,t}]\right\|^2\right] + \left\|\frac{1}{n}\sum_{i\in[n]}\mathbb{E}[g_{i,t}]\right\|^2\right]$$

$$= F(x_{t-1}) + \frac{\eta}{2}\|\nabla F(x_{t-1}) - \frac{1}{n}\sum_{i\in[n]}\mathbb{E}[g_{i,t}]\|^2 - \frac{\eta}{2}\|\nabla F(x_{t-1})\|^2 - \frac{\eta}{2}\left\|\frac{1}{n}\sum_{i\in[n]}\mathbb{E}[g_{i,t}]\right\|^2$$

$$+ \frac{L\eta^2}{2}\mathbb{E}\left[\left\|\frac{1}{n}\sum_{i\in[n]}g_{i,t} - \frac{1}{n}\sum_{i\in[n]}\mathbb{E}[g_{i,t}]\right\|^2\right] + \frac{L\eta^2}{2}\left\|\frac{1}{n}\sum_{i\in[n]}\mathbb{E}[g_{i,t}]\right\|^2$$

$$\leq F(x_{t-1}) - \frac{\eta}{2}\|\nabla F(x_{t-1})\|^2$$

$$+ \frac{\eta}{2}\left\|\nabla F(x_{t-1}) - \frac{1}{n}\sum_{i\in[n]}\mathbb{E}[g_{i,t}]\right\|^2 + \frac{L\eta^2}{2}\mathbb{E}\left[\left\|\frac{1}{n}\sum_{i\in[n]}g_{i,t} - \frac{1}{n}\sum_{i\in[n]}\mathbb{E}[g_{i,t}]\right\|^2\right]. \qquad \triangleright \eta \leq \frac{1}{L}$$

$\square$

Using Lemma 1, we establish the error bound of the convergence of vanilla SAPipe, Algorithm 2 (SAPipe-DC) Algorithm 3 (SAPipe-WP).

**Theorem 2.** *Under Assumption 1, 2, 5, and 6, taking $\eta \leq \frac{1}{L}$, after $T$ iterations, for vanilla SAPipe without DC or WP, we have the following error bound:*

$$\frac{1}{T}\sum_{t=1}^{T}\mathbb{E}\left[\|\nabla F(x_{t-1})\|^2\right] \leq \frac{2R_0}{\eta T} + \kappa Err_0 + Var_0,$$

*where $Err_0 = L^2\eta^2 V_2$, $Var_0 = \frac{L\eta V_1}{n}$.*

*Proof.* For vanilla SAPipe, we have $\mathbb{E}[g_{i,t}] = \nabla F_i(x_{t-2})$ if the gradient has one-step staleness.

Thus, for the variance term, we have

$$
\mathbb{E}\left[\left\|\frac{1}{n}\sum_{i\in[n]}g_{i,t}-\frac{1}{n}\sum_{i\in[n]}\mathbb{E}[g_{i,t}]\right\|^2\right]
$$

$$
=\frac{1}{n^2}\sum_{i\in[n]}\mathbb{E}\left[\|g_{i,t}-\mathbb{E}[g_{i,t}]\|^2\right]
$$

$$
=\frac{1}{n^2}\sum_{i\in[n]}\mathbb{E}\left[\|g_{i,t}-\nabla f_i(x_{t-2})\|^2\right]
$$

$$
\leq\frac{V_1}{n}.
$$

For the error of the gradient estimation, we have

$$
\left\|\nabla F(x_{t-1})-\frac{1}{n}\sum_{i\in[n]}\mathbb{E}[g_{i,t}]\right\|^2
$$

$$
=\left\|\nabla F(x_{t-1})-\frac{1}{n}\sum_{i\in[n]}\nabla F_i(x_{t-2})\right\|^2
$$

$$
=\|\nabla F(x_{t-1})-\nabla F(x_{t-2})\|^2
$$

$$
\leq L^2\|x_{t-1}-x_{t-2}\|^2 \qquad\qquad \triangleright\ L\text{-smoothness}
$$

$$
=L^2\left\|\eta\frac{1}{n}\sum_{i\in[n]}g_{i,t-1}\right\|^2
$$

$$
=L^2\eta^2\left\|\frac{1}{n}\sum_{i\in[n]}\nabla f_i(x_{t-3})\right\|^2.
$$

Putting all the ingredients together, we have

$$
\mathbb{E}[F(x_t)]
$$

$$
\leq F(x_{t-1})-\frac{\eta}{2}\|\nabla F(x_{t-1})\|^2
$$

$$
+\frac{\eta}{2}\left\|\nabla F(x_{t-1})-\frac{1}{n}\sum_{i\in[n]}\mathbb{E}[g_{i,t}]\right\|^2+\frac{L\eta^2}{2}\mathbb{E}\left[\left\|\frac{1}{n}\sum_{i\in[n]}g_{i,t}-\frac{1}{n}\sum_{i\in[n]}\mathbb{E}[g_{i,t}]\right\|^2\right]
$$

$$
\leq F(x_{t-1})-\frac{\eta}{2}\|\nabla F(x_{t-1})\|^2+\frac{\kappa\eta}{2}L^2\eta^2\left\|\frac{1}{n}\sum_{i\in[n]}\nabla f_i(x_{t-3})\right\|^2+\frac{L\eta^2}{2}\frac{V_1}{n}.
$$

By re-arranging the terms, we have

$$
\|\nabla F(x_{t-1})\|^2\leq\frac{2\mathbb{E}[F(x_{t-1})-F(x_t)]}{\eta}+\kappa L^2\eta^2\left\|\frac{1}{n}\sum_{i\in[n]}\nabla f_i(x_{t-3})\right\|^2+\frac{L\eta V_1}{n}.
$$

By telescoping and taking total expectation, after $T$ iterations, we have

$$
\frac{1}{T}\sum_{t=1}^{T}\mathbb{E}\left[\|\nabla F(x_{t-1})\|^2\right]
$$

$$\leq \frac{2\mathbb{E}[F(x_0) - F(x_*)]}{\eta T} + L^2\eta^2 \frac{1}{T}\sum_{t=3}^{T} \mathbb{E}\left\|\frac{1}{n}\sum_{i\in[n]}\nabla f_i(x_{t-3})\right\|^2 + \frac{L\eta V_1}{n}$$

$$\leq \frac{2\mathbb{E}[F(x_0) - F(x_*)]}{\eta T} + \kappa L^2\eta^2 V_2 + \frac{L\eta V_1}{n}. \qquad \rhd \text{ Assumption 2}$$

$\square$

**Theorem 3.** *Under Assumption 1, 2, 4, 5, and 6, taking a small enough $\lambda$ so that $\lambda\eta V_2 \leq 1 - \frac{1}{\sqrt{2}}$, and $\eta \leq \frac{1}{L}$, after $T$ iterations, for Algorithm 2 (SAPipe-DC), we have the following error bound:*

$$\frac{1}{T}\sum_{t=1}^{T}\mathbb{E}\left[\|\nabla F(x_{t-1})\|^2\right] \leq \frac{2R_0}{\eta T} + \kappa Err_{DC} + (1+\kappa)Var_0,$$

*where $Err_{DC} = 128\eta^4 M^2 V_2^2 + 32\eta^2(1-\lambda)^2 L^2 V_2 + 32\eta^2\lambda^2\Delta^2 V_2 + 16\eta^3\lambda^2 LV_2^3$.*

*Proof.* Thus, for the variance term, we have

$$\mathbb{E}\left[\left\|g_t^{DC} - \mathbb{E}[g_t^{DC}]\right\|^2\right]$$

$$= \mathbb{E}\left[\left\|g_t + \lambda g_t g_t^\top(x_{t-1}-x_{t-2}) - \mathbb{E}[g_t + \lambda g_t g_t^\top(x_{t-1}-x_{t-2})]\right\|^2\right]$$

$$= \mathbb{E}\left[\left\|g_t - \mathbb{E}[g_t] + \lambda g_t g_t^\top(x_{t-1}-x_{t-2}) - \mathbb{E}[\lambda g_t g_t^\top(x_{t-1}-x_{t-2})]\right\|^2\right]$$

$$\leq 2\mathbb{E}\left[\|g_t - \mathbb{E}[g_t]\|^2\right] + 2\mathbb{E}\left[\left\|\lambda g_t g_t^\top(x_{t-1}-x_{t-2}) - \mathbb{E}[\lambda g_t g_t^\top(x_{t-1}-x_{t-2})]\right\|^2\right]$$

$$\leq \frac{2V_1}{n} + 2\mathbb{E}\left[\left\|\lambda g_t g_t^\top(x_{t-1}-x_{t-2}) - \mathbb{E}[\lambda g_t g_t^\top(x_{t-1}-x_{t-2})]\right\|^2\right]$$

$$\leq \frac{2V_1}{n} + 2\lambda^2\mathbb{E}\left[\left\|g_t g_t^\top(x_{t-1}-x_{t-2})\right\|^2\right]$$

$$\leq \frac{2V_1}{n} + 2\lambda^2\mathbb{E}\left[\|g_t\|^2\|g_t\|^2\|x_{t-1}-x_{t-2}\|^2\right]$$

$$\leq \frac{2V_1}{n} + 2\lambda^2 V_2^2\|x_{t-1}-x_{t-2}\|^2.$$

For the error of the gradient estimation, we have

$$\left\|\nabla F(x_{t-1}) - \mathbb{E}[g_t^{DC}]\right\|^2$$

$$= \left\|\nabla F(x_{t-1}) - \mathbb{E}[g_t + \lambda g_t g_t^\top(x_{t-1}-x_{t-2})]\right\|^2$$

$$= \left\|\nabla F(x_{t-1}) - \mathbb{E}[g_t + \nabla^2 f(x_{t-2})(x_{t-1}-x_{t-2})] + \mathbb{E}[g_t + \nabla^2 f(x_{t-2})(x_{t-1}-x_{t-2})] - \mathbb{E}[g_t + \lambda g_t g_t^\top(x_{t-1}-x_{t-2})]\right\|^2$$

$$\leq 2\left\|\nabla F(x_{t-1}) - \mathbb{E}[g_t + \nabla^2 f(x_{t-2})(x_{t-1}-x_{t-2})]\right\|^2$$
$$+ 2\left\|\mathbb{E}[g_t + \nabla^2 f(x_{t-2})(x_{t-1}-x_{t-2})] - \mathbb{E}[g_t + \lambda g_t g_t^\top(x_{t-1}-x_{t-2})]\right\|^2$$

$$\leq 2M^2\|x_{t-1}-x_{t-2}\|^4 + 2\left\|\mathbb{E}[\nabla^2 f(x_{t-2})(x_{t-1}-x_{t-2})] - \mathbb{E}[\lambda g_t g_t^\top(x_{t-1}-x_{t-2})]\right\|^2$$
$$\rhd \text{ Assumption 4}$$

$$\leq 2M^2\|x_{t-1}-x_{t-2}\|^4$$
$$+ 2\left\|\mathbb{E}[(1-\lambda)\nabla^2 f(x_{t-2})(x_{t-1}-x_{t-2}) + \lambda(\nabla^2 f(x_{t-2})(x_{t-1}-x_{t-2}) - g_t g_t^\top(x_{t-1}-x_{t-2}))]\right\|^2$$

$$\leq 2M^2\|x_{t-1}-x_{t-2}\|^4 + 4(1-\lambda)^2\left\|\mathbb{E}[\nabla^2 f(x_{t-2})(x_{t-1}-x_{t-2})]\right\|^2$$
$$+ 4\lambda^2\left\|\mathbb{E}[\nabla^2 f(x_{t-2})(x_{t-1}-x_{t-2}) - g_t g_t^\top(x_{t-1}-x_{t-2})]\right\|^2$$

$$\leq 2M^2\|x_{t-1}-x_{t-2}\|^4 + 4(1-\lambda)^2 L^2\|x_{t-1}-x_{t-2}\|^2$$
$$+ 4\lambda^2\left\|\mathbb{E}[\nabla^2 f(x_{t-2})(x_{t-1}-x_{t-2}) - g_t g_t^\top(x_{t-1}-x_{t-2})]\right\|^2$$

$$\leq 2M^2\|x_{t-1}-x_{t-2}\|^4 + 4(1-\lambda)^2 L^2\|x_{t-1}-x_{t-2}\|^2 + 4\lambda^2\Delta^2\|x_{t-1}-x_{t-2}\|^2.$$
$$\rhd \text{ Assumption 4}$$

To finish the proof, we need to establish the upper bound of $\|x_{t-1} - x_{t-2}\|$. It is easy to check that the term $a_t = \|x_t - x_{t-1}\|$ has the following recursive inequality:

$$\|x_t - x_{t-1}\|$$
$$= \|\eta(g_t + \lambda g_t g_t^\top (x_{t-1} - x_{t-2}))\|$$
$$\leq \eta\|g_t\| + \eta\lambda\|g_t g_t^\top (x_{t-1} - x_{t-2})\| \qquad \triangleright \text{ triangle inequality}$$
$$\leq \eta\|g_t\| + \eta\lambda\|g_t\|^2\|x_{t-1} - x_{t-2}\|$$
$$\leq \eta\sqrt{V_2} + \eta\lambda V_2\|x_{t-1} - x_{t-2}\|.$$

Or, $a_t \leq \eta\sqrt{V_2} + \eta\lambda V_2 a_{t-1}$. Note that $a_1 = \|x_1 - x_0\| = \|\eta g_1\| \leq \eta\sqrt{V_2}$. Thus, we have $\forall t \geq 2$:

$$\|x_t - x_{t-1}\| \leq \eta\sqrt{V_2}\sum_{\tau=0}^{t-2}(\lambda\eta V_2)^\tau + \eta\sqrt{V_2}(\lambda\eta V_2)^{t-1} \leq \frac{2\eta\sqrt{V_2}}{1 - \lambda\eta V_2} \leq 2\sqrt{2}\eta\sqrt{V_2}.$$

Putting all the ingredients together, we have

$$\mathbb{E}[F(x_t)]$$
$$\leq F(x_{t-1}) - \frac{\eta}{2}\|\nabla F(x_{t-1})\|^2$$
$$+ \frac{\kappa\eta}{2}[2M^2\|x_{t-1} - x_{t-2}\|^4 + 4(1-\lambda)^2 L^2\|x_{t-1} - x_{t-2}\|^2 + 4\lambda^2\Delta^2\|x_{t-1} - x_{t-2}\|^2]$$
$$+ \frac{\kappa L\eta^2}{2}[\frac{2V_1}{n} + 2\lambda^2 V_2^2\|x_{t-1} - x_{t-2}\|^2] + \frac{(1-\kappa)L\eta^2 V_1}{2n}$$
$$\leq F(x_{t-1}) - \frac{\eta}{2}\|\nabla F(x_{t-1})\|^2$$
$$+ \kappa(64\eta^5 M^2 V_2^2 + 16\eta^3(1-\lambda)^2 L^2 V_2 + 16\eta^3\lambda^2\Delta^2 V_2 + 8\eta^4\lambda^2 LV_2^3) + \frac{(1+\kappa)\eta^2 LV_1}{2n}.$$

By re-arranging the terms, we have

$$\|\nabla F(x_{t-1})\|^2$$
$$+ \kappa(128\eta^4 M^2 V_2^2 + 32\eta^2(1-\lambda)^2 L^2 V_2 + 32\eta^2\lambda^2\Delta^2 V_2 + 16\eta^3\lambda^2 LV_2^3) + \frac{(1+\kappa)\eta LV_1}{n}.$$

By telescoping and taking total expectation, after $T$ iterations, we have

$$\frac{1}{T}\sum_{t=1}^T \mathbb{E}\left[\|\nabla F(x_{t-1})\|^2\right]$$
$$\leq \frac{2\mathbb{E}[F(x_0) - F(x_*)]}{\eta T}$$
$$+ \kappa(128\eta^4 M^2 V_2^2 + 32\eta^2(1-\lambda)^2 L^2 V_2 + 32\eta^2\lambda^2\Delta^2 V_2 + 16\eta^3\lambda^2 LV_2^3) + \frac{(1+\kappa)\eta LV_1}{n}.$$

$$\square$$

**Theorem 4.** *Under Assumption 1, 2, 3, 5, and 6, taking $\eta \leq \frac{1}{L}$, after $T$ iterations, for Algorithm 3 (SAPipe-WP) with Option 1, we have the following error bound:*

$$\frac{1}{T}\sum_{t=1}^T \mathbb{E}\left[\|\nabla F(x_{t-1})\|^2\right] \leq \frac{2R_0}{\eta T} + \kappa Err_{WP-1} + Var_0,$$

*where $Err_{WP-1} = L^2\eta^2\left[\left(\frac{\rho}{n} - 1\right)V_2 + \left(3 - \frac{\rho}{n}\right)V_1\right]$.*

*Proof.* For SAPipe-WP with Option 1, we have $\mathbb{E}[g_{i,t}] = \nabla F_i(\tilde{x}_{i,t-1})$.

Thus, for the variance term, since $g_{i,t}$ is a valid gradient, similar to Theorem 2, we have

$$\mathbb{E}\left[\left\|\frac{1}{n}\sum_{i\in[n]} g_{i,t} - \frac{1}{n}\sum_{i\in[n]}\mathbb{E}[g_{i,t}]\right\|^2\right]$$

$$= \frac{1}{n^2} \sum_{i \in [n]} \mathbb{E}\left[\|g_{i,t} - \mathbb{E}[g_{i,t}]\|^2\right]$$

$$\leq \frac{V_1}{n}.$$

For the error of the gradient estimation, we have

$$\left\|\nabla F(x_{t-1}) - \frac{1}{n} \sum_{i \in [n]} \mathbb{E}[g_{i,t}]\right\|^2$$

$$= \left\|\nabla F(x_{t-1}) - \frac{1}{n} \sum_{i \in [n]} \nabla F_i(\tilde{x}_{i,t-1})\right\|^2$$

$$= \left\|\frac{1}{n} \sum_{i \in [n]} \nabla F_i(x_{t-1}) - \frac{1}{n} \sum_{i \in [n]} \nabla F_i(\tilde{x}_{i,t-1})\right\|^2$$

$$\leq \frac{1}{n} \sum_{i \in [n]} \|\nabla F_i(x_{t-1}) - \nabla F_i(\tilde{x}_{i,t-1})\|^2$$

$$\leq L^2 \frac{1}{n} \sum_{i \in [n]} \|x_{t-1} - \tilde{x}_{i,t-1}\|^2 \qquad \triangleright L\text{-smoothness}$$

$$= L^2 \frac{1}{n} \sum_{i \in [n]} \left\|\eta \frac{1}{n} \sum_{j \in [n]} (g_{j,t-1}) - \eta g_{i,t-1}\right\|^2$$

$$= \frac{L^2 \eta^2}{n} \sum_{i \in [n]} \left\|\frac{1}{n} \sum_{j \in [n]} (g_{j,t-1}) - g_{i,t-1}\right\|^2$$

Putting all the ingredients together, we have

$$\mathbb{E}[F(x_t)]$$

$$\leq F(x_{t-1}) - \frac{\eta}{2}\|\nabla F(x_{t-1})\|^2$$

$$+ \frac{\eta}{2}\left\|\nabla F(x_{t-1}) - \frac{1}{n} \sum_{i \in [n]} \mathbb{E}[g_{i,t}]\right\|^2 + \frac{L\eta^2}{2}\mathbb{E}\left[\left\|\frac{1}{n} \sum_{i \in [n]} g_{i,t} - \frac{1}{n} \sum_{i \in [n]} \mathbb{E}[g_{i,t}]\right\|^2\right]$$

$$\leq F(x_{t-1}) - \frac{\eta}{2}\|\nabla F(x_{t-1})\|^2 + \frac{\kappa\eta}{2}\frac{L^2\eta^2}{n} \sum_{i \in [n]} \left\|\frac{1}{n} \sum_{j \in [n]} (g_{j,t-1}) - g_{i,t-1}\right\|^2 + \frac{L\eta^2}{2}\frac{V_1}{n}.$$

By re-arranging the terms, we have

$$\|\nabla F(x_{t-1})\|^2 \leq \frac{2\mathbb{E}[F(x_{t-1}) - F(x_t)]}{\eta} + \frac{\kappa L^2\eta^2}{n} \sum_{i \in [n]} \left\|\frac{1}{n} \sum_{j \in [n]} (g_{j,t-1}) - g_{i,t-1}\right\|^2 + \frac{L\eta V_1}{n}.$$

By telescoping and taking total expectation, after $T$ iterations, we have

$$\frac{1}{T} \sum_{t=1}^{T} \mathbb{E}\left[\|\nabla F(x_{t-1})\|^2\right]$$

$$\leq \frac{2\mathbb{E}[F(x_0) - F(x_*)]}{\eta T} + \frac{1}{T} \sum_{t=3}^{T} L^2\eta^2 \frac{1}{n} \sum_{i \in [n]} \mathbb{E}\left\|\frac{1}{n} \sum_{j \in [n]} (g_{j,t-1}) - g_{i,t-1}\right\|^2 + \frac{L\eta V_1}{n}.$$

To finish the proof, we establish the following upper bound for $\forall i, j \in [n], i \neq j$

$$\frac{1}{n} \sum_{i \in [n]} \mathbb{E} \left\| \frac{1}{n} \sum_{j \in [n]} (g_{j,t-1}) - g_{i,t-1} \right\|^2$$

$$= \frac{1}{n} \sum_{i \in [n]} \mathbb{E} \left\| \left( \frac{1}{n} \sum_{j \in [n]} \nabla f_j(\tilde{x}_{t-2}) \right) - \nabla f_i(\tilde{x}_{t-2}) \right\|^2$$

$$= \frac{1}{n} \sum_{i \in [n]} \mathbb{E} \left\| \left( \frac{1}{n} \sum_{j \in [n]} \nabla f_j(\tilde{x}_{t-2}) \right) - \nabla f_i(\tilde{x}_{t-2}) \pm (\nabla F(\tilde{x}_{t-2}) - \nabla F_i(\tilde{x}_{t-2})) \right\|^2$$

$$= \frac{1}{n} \sum_{i \in [n]} \mathbb{E} \left\| \left( \frac{1}{n} \sum_{j \in [n]} \nabla f_j(\tilde{x}_{t-2}) \right) - \nabla F(\tilde{x}_{t-2}) \right\|^2$$

$$+ \frac{1}{n} \sum_{i \in [n]} \mathbb{E} \left\| \nabla F_i(\tilde{x}_{t-2}) - \nabla f_i(\tilde{x}_{t-2}) \right\|^2$$

$$+ \frac{1}{n} \sum_{i \in [n]} \mathbb{E} \left\| \nabla F(\tilde{x}_{t-2}) - \nabla F_i(\tilde{x}_{t-2}) \right\|^2$$

$$\leq \frac{1}{n} \sum_{i \in [n]} \mathbb{E} \left\| \nabla F(\tilde{x}_{t-2}) - \nabla F_i(\tilde{x}_{t-2}) \right\|^2 + 2V_1$$

$$\leq \left( \frac{\rho}{n} - 1 \right) \mathbb{E} \left\| \nabla F(\tilde{x}_{t-2}) \right\|^2 + 2V_1 \qquad \triangleright \text{Assumption 3}$$

$$\leq \left( \frac{\rho}{n} - 1 \right) V_1' + 2V_1 \qquad \triangleright \text{Assumption 2}$$

$$= \left( \frac{\rho}{n} - 1 \right) (V_2 - V_1) + 2V_1 \qquad \triangleright \text{Assumption 2, } V_2 = V_1 + V_1'$$

$$\leq \left( \frac{\rho}{n} - 1 \right) V_2 + \left( 3 - \frac{\rho}{n} \right) V_1.$$

Finally, putting all the ingredients above together, we have

$$\frac{1}{T} \sum_{t=1}^{T} \mathbb{E} \left[ \|\nabla F(x_{t-1})\|^2 \right]$$

$$\leq \frac{2\mathbb{E}[F(x_0) - F(x_*)]}{\eta T} + \kappa L^2 \eta^2 \left[ \left( \frac{\rho}{n} - 1 \right) V_2 + \left( 3 - \frac{\rho}{n} \right) V_1 \right] + \frac{L\eta V_1}{n}.$$

$\square$

**Theorem 5.** *Under Assumption 1, 2, 5, and 6, taking $\eta \leq \frac{1}{L}$, after $T$ iterations, for Algorithm 3 (SAPipe-WP) with Option 2, we have the following error bound:*

$$\frac{1}{T} \sum_{t=1}^{T} \mathbb{E} \left[ \|\nabla F(x_{t-1})\|^2 \right] \leq \frac{2R_0}{\eta T} + \kappa Err_{WP-2} + Var_0,$$

*where $Err_{WP-2} = \frac{L^2 \eta^2}{1 - 2L^2 \eta^2} (2V_1 + 2L^2 \eta^2 V_2)$.*

*Proof.* For SAPipe-WP with Option 2, we have $\mathbb{E}[g_{i,t}] = \nabla F_i(\tilde{x}_{t-1})$.

Thus, for the variance term, since $g_{i,t}$ is a valid gradient, similar to Theorem 2, we have

$$\mathbb{E} \left[ \left\| \frac{1}{n} \sum_{i \in [n]} g_{i,t} - \frac{1}{n} \sum_{i \in [n]} \mathbb{E}[g_{i,t}] \right\|^2 \right]$$

$$= \frac{1}{n^2} \sum_{i \in [n]} \mathbb{E}\left[\|g_{i,t} - \mathbb{E}[g_{i,t}]\|^2\right]$$

$$\leq \frac{V_1}{n}.$$

For the error of the gradient estimation, we have

$$\left\|\nabla F(x_{t-1}) - \frac{1}{n} \sum_{i \in [n]} \mathbb{E}[g_{i,t}]\right\|^2$$

$$= \left\|\nabla F(x_{t-1}) - \frac{1}{n} \sum_{i \in [n]} \nabla F_i(\tilde{x}_{t-1})\right\|^2$$

$$= \|\nabla F(x_{t-1}) - \nabla F(\tilde{x}_{t-1})\|^2$$

$$\leq L^2 \|x_{t-1} - \tilde{x}_{t-1}\|^2 \qquad\qquad\qquad\qquad\qquad \triangleright L\text{-smoothness}$$

$$= L^2 \left\|\eta \frac{1}{n} \sum_{i \in [n]} (g_{i,t-1} - g_{i,t-2})\right\|^2$$

$$= L^2 \eta^2 \left\|\frac{1}{n} \sum_{i \in [n]} (\nabla f(\tilde{x}_{t-2}; z_{i,t-2}) - \nabla f(\tilde{x}_{t-3}; z_{i,t-3}))\right\|^2.$$

Putting all the ingredients together, we have

$$\mathbb{E}[F(x_t)]$$

$$\leq F(x_{t-1}) - \frac{\eta}{2}\|\nabla F(x_{t-1})\|^2$$

$$+ \frac{\eta}{2}\left\|\nabla F(x_{t-1}) - \frac{1}{n} \sum_{i \in [n]} \mathbb{E}[g_{i,t}]\right\|^2 + \frac{L\eta^2}{2}\mathbb{E}\left[\left\|\frac{1}{n} \sum_{i \in [n]} g_{i,t} - \frac{1}{n} \sum_{i \in [n]} \mathbb{E}[g_{i,t}]\right\|^2\right]$$

$$\leq F(x_{t-1}) - \frac{\eta}{2}\|\nabla F(x_{t-1})\|^2 + \frac{\kappa\eta}{2} L^2 \eta^2 \left\|\frac{1}{n} \sum_{i \in [n]} (\nabla f(\tilde{x}_{t-2}; z_{i,t-2}) - \nabla f(\tilde{x}_{t-3}; z_{i,t-3}))\right\|^2$$

$$+ \frac{L\eta^2}{2}\frac{V_1}{n}.$$

By re-arranging the terms, we have

$$\|\nabla F(x_{t-1})\|^2$$

$$\leq \frac{2\mathbb{E}[F(x_{t-1}) - F(x_t)]}{\eta} + \kappa L^2 \eta^2 \left\|\frac{1}{n} \sum_{i \in [n]} (\nabla f(\tilde{x}_{t-2}; z_{i,t-2}) - \nabla f(\tilde{x}_{t-3}; z_{i,t-3}))\right\|^2$$

$$+ \frac{L\eta V_1}{n}.$$

By telescoping and taking total expectation, after $T$ iterations, we have

$$\frac{1}{T} \sum_{t=1}^{T} \mathbb{E}\left[\|\nabla F(x_{t-1})\|^2\right]$$

$$\leq \frac{2\mathbb{E}[F(x_0) - F(x_*)]}{\eta T} + \frac{1}{T} \sum_{t=3}^{T} \kappa L^2 \eta^2 \mathbb{E}\left\|\frac{1}{n} \sum_{i \in [n]} (\nabla f(\tilde{x}_{t-2}; z_{i,t-2}) - \nabla f(\tilde{x}_{t-3}; z_{i,t-3}))\right\|^2$$

$$+ \frac{L\eta V_1}{n}.$$

To finish the proof, we establish the following upper bound

$$\mathbb{E} \left\| \frac{1}{n} \sum_{i \in [n]} (\nabla f(\tilde{x}_{t-2}; z_{i,t-2}) - \nabla f(\tilde{x}_{t-3}; z_{i,t-3})) \right\|^2$$

$$\leq \mathbb{E} \left\| \frac{1}{n} \sum_{i \in [n]} [(\nabla f(\tilde{x}_{t-2}; z_{i,t-2}) - \nabla F(\tilde{x}_{t-2})) + (\nabla F(\tilde{x}_{t-3}) - \nabla f(\tilde{x}_{t-3}; z_{i,t-3})) + (\nabla F(\tilde{x}_{t-2}) - \nabla F(\tilde{x}_{t-3}))] \right\|^2$$

$$\leq \mathbb{E} \left\| \frac{1}{n} \sum_{i \in [n]} (\nabla f(\tilde{x}_{t-2}; z_{i,t-2}) - \nabla F(\tilde{x}_{t-2})) \right\|^2 + \mathbb{E} \left\| \frac{1}{n} \sum_{i \in [n]} (\nabla F(\tilde{x}_{t-3}) - \nabla f(\tilde{x}_{t-3}; z_{i,t-3})) \right\|^2$$

$$+ \mathbb{E} \left\| \frac{1}{n} \sum_{i \in [n]} (\nabla F(\tilde{x}_{t-2}) - \nabla F(\tilde{x}_{t-3})) \right\|^2$$

$$\leq \mathbb{E} \left\| \nabla F(\tilde{x}_{t-2}) - \nabla F(\tilde{x}_{t-3}) \right\|^2 + 2V_1$$

$$\leq L^2 \mathbb{E} \left\| \tilde{x}_{t-2} - \tilde{x}_{t-3} \right\|^2 + 2V_1$$

$$= L^2 \mathbb{E} \left\| x_{t-3} - x_{t-4} - \eta(g_{t-3} - g_{t-4}) \right\|^2 + 2V_1$$

$$\leq 2L^2 \mathbb{E} \left\| x_{t-3} - x_{t-4} \right\|^2 + 2L^2 \eta^2 \mathbb{E} \left\| g_{t-3} - g_{t-4} \right\|^2 + 2V_1$$

$$\leq 2L^2 \mathbb{E} \left\| \eta g_{t-3} \right\|^2 + 2L^2 \eta^2 \mathbb{E} \left\| g_{t-3} - g_{t-4} \right\|^2 + 2V_1$$

$$\leq 2L^2 \eta^2 \mathbb{E} \left\| g_{t-3} - g_{t-4} \right\|^2 + 2V_1 + 2L^2 \eta^2 V_2,$$

where $g_t = \frac{1}{n} \sum_{i \in [n]} g_{i,t}$.

Thus, we have the following recursive upper bound

$$\mathbb{E} \left\| g_{t-1} - g_{t-2} \right\|^2 \leq 2L^2 \eta^2 \mathbb{E} \left\| g_{t-3} - g_{t-4} \right\|^2 + 2V_1 + 2L^2 \eta^2 V_2,$$

and at the very beginning, there is no staleness, thus $g_1 - g_2 = 0$.

Thus, using $\eta \leq \frac{1}{2L}$, we have

$$\mathbb{E} \left\| g_{t-1} - g_{t-2} \right\|^2 \leq \frac{1}{1 - 2L^2 \eta^2} (2V_1 + 2L^2 \eta^2 V_2).$$

Finally, putting all the ingredients together, we have

$$\frac{1}{T} \sum_{t=1}^{T} \mathbb{E} \left[ \|\nabla F(x_{t-1})\|^2 \right] \leq \frac{2\mathbb{E}[F(x_0) - F(x_*)]}{\eta T} + \frac{\kappa L^2 \eta^2}{1 - 2L^2 \eta^2} (2V_1 + 2L^2 \eta^2 V_2) + \frac{L\eta V_1}{n}.$$

$\square$

**Theorem 6.** *Under Assumption 1, 2, 4, 5, and 6, taking $\eta \leq \frac{1}{L}$, after $T$ iterations, for Algorithm 3 (SAPipe-WP) with Option 3, we have the following error bound:*

$$\frac{1}{T} \sum_{t=1}^{T} \mathbb{E} \left[ \|\nabla F(x_{t-1})\|^2 \right] \leq \frac{2R_0}{\eta T} + \kappa Err_{WP-3} + Var_0,$$

*where $Err_{WP-3} = \eta^2 L^2 [8\frac{V_1}{n} + 4\eta^4 M^2 V_2^2 + 2\eta^2 V_2 (L^2(1-\lambda)^2 + \lambda^2 \Delta^2)]$.*

*Proof.* Note that similar to the other options, in this case we still have $g_{i,t}$ as a valid stochastic gradient. Thus, for the variance term, we have

$$\mathbb{E} \left[ \left\| \frac{1}{n} \sum_{i \in [n]} g_{i,t} - \frac{1}{n} \sum_{i \in [n]} \mathbb{E}[g_{i,t}] \right\|^2 \right]$$

$$= \frac{1}{n^2} \sum_{i \in [n]} \mathbb{E} \left[ \|g_{i,t} - \mathbb{E}[g_{i,t}]\|^2 \right]$$

$$\leq \frac{V_1}{n}.$$

For the error of the gradient estimation, we have

$$\left\| \nabla F(x_{t-1}) - \frac{1}{n} \sum_{i \in [n]} \mathbb{E}[g_{i,t}] \right\|^2$$

$$= \left\| \nabla F(x_{t-1}) - \frac{1}{n} \sum_{i \in [n]} \nabla F_i(\tilde{x}_{i,t-1}) \right\|^2$$

$$= \left\| \frac{1}{n} \sum_{i \in [n]} \nabla F_i(x_{t-1}) - \frac{1}{n} \sum_{i \in [n]} \nabla F_i(\tilde{x}_{i,t-1}) \right\|^2$$

$$\leq \frac{1}{n} \sum_{i \in [n]} \left\| \nabla F_i(x_{t-1}) - \nabla F_i(\tilde{x}_{i,t-1}) \right\|^2$$

$$\leq L^2 \frac{1}{n} \sum_{i \in [n]} \| x_{t-1} - \tilde{x}_{i,t-1} \|^2 \qquad\qquad \triangleright \text{ } L\text{-smoothness}$$

$$= L^2 \frac{1}{n} \sum_{i \in [n]} \left\| \eta \frac{1}{n} \sum_{j \in [n]} (g_{j,t-1}) - \eta \left( DC(g_{t-2} - \frac{1}{n} g_{i,t-2}, \Delta x_{t-2}) + \frac{1}{n} g_{i,t-1} \right) \right\|^2$$

$$= \eta^2 L^2 \frac{1}{n} \sum_{i \in [n]} \left\| \frac{1}{n} \sum_{j \neq i} (g_{j,t-1}) - DC(g_{t-2} - \frac{1}{n} g_{i,t-2}, \Delta x_{t-2}) \right\|^2$$

$$= \eta^2 L^2 \frac{1}{n} \sum_{i \in [n]} \left\| \frac{1}{n} \sum_{j \neq i} (g_{j,t-1}) - DC \left( \frac{1}{n} \sum_{j \neq i} (g_{j,t-2}), \Delta x_{t-2} \right) \right\|^2$$

$$= \eta^2 L^2 \frac{1}{n} \sum_{i \in [n]} \left\| \frac{1}{n} \sum_{j \neq i} (g_{j,t-1}) - \left[ \frac{1}{n} \sum_{j \neq i} (g_{j,t-2}) + \lambda \left( \frac{1}{n} \sum_{j \neq i} (g_{j,t-2}) \right) \left( \frac{1}{n} \sum_{j \neq i} (g_{j,t-2}) \right)^{\top} (x_{t-2} - x_{t-3}) \right] \right\|^2$$

$$= \underbrace{2\eta^2 L^2 \frac{1}{n} \sum_{i \in [n]} \left\| \frac{1}{n} \sum_{j \neq i} (g_{j,t-1}) - \left[ \frac{1}{n} \sum_{j \neq i} (g_{j,t-2} + \nabla^2 f(x_{t-3}, z_{j,t-3})(x_{t-2} - x_{t-3})) \right] \right\|^2}_{\text{①}}$$

$$+ \underbrace{2\eta^2 L^2 \frac{1}{n} \sum_{i \in [n]} \left\| \frac{1}{n} \sum_{j \neq i} \nabla^2 f(x_{t-3}, z_{j,t-3})(x_{t-2} - x_{t-3}) - \lambda \left( \frac{1}{n} \sum_{j \neq i} (g_{j,t-2}) \right) \left( \frac{1}{n} \sum_{j \neq i} (g_{j,t-2}) \right)^{\top} (x_{t-2} - x_{t-3}) \right\|^2}_{\text{②}}.$$

For ①, we have

①

$$= 2\eta^2 L^2 \frac{1}{n} \sum_{i \in [n]} \left\| \frac{1}{n} \sum_{j \neq i} (g_{j,t-1}) - \left[ \frac{1}{n} \sum_{j \neq i} (g_{j,t-2} + \nabla^2 f(x_{t-3}, z_{j,t-3})(x_{t-2} - x_{t-3})) \right] \right\|^2$$

$$= 2\eta^2 L^2 \frac{1}{n} \sum_{i \in [n]} \left\| \frac{1}{n} \sum_{j \neq i} \nabla f(x_{t-2}, z_{j,t-2}) - \left[ \frac{1}{n} \sum_{j \neq i} \nabla f(x_{t-3}, z_{j,t-3}) + \nabla^2 f(x_{t-3}, z_{j,t-3})(x_{t-2} - x_{t-3}) \right] \right\|^2$$

$$\leq 4\eta^2 L^2 \frac{1}{n} \sum_{i \in [n]} \left\| \frac{1}{n} \sum_{j \neq i} \nabla f(x_{t-2}, z_{j,t-2}) - \frac{1}{n} \sum_{j \neq i} \nabla f(x_{t-2}, z_{j,t-3}) \right\|^2$$

$$+ 4\eta^2 L^2 \frac{1}{n} \sum_{i \in [n]} \left\| \frac{1}{n} \sum_{j \neq i} \nabla f(x_{t-2}, z_{j,t-3}) - \left[ \frac{1}{n} \sum_{j \neq i} \nabla f(x_{t-3}, z_{j,t-3}) + \nabla^2 f(x_{t-3}, z_{j,t-3})(x_{t-2} - x_{t-3}) \right] \right\|^2$$

$$\leq 4\eta^2 L^2 \frac{1}{n} \sum_{i \in [n]} \left\| \frac{1}{n} \sum_{j \neq i} \nabla f(x_{t-2}, z_{j,t-2}) - \frac{1}{n} \sum_{j \neq i} \nabla f(x_{t-2}, z_{j,t-3}) \right\|^2 + 4\eta^2 L^2 M^2 \|x_{t-2} - x_{t-3}\|^4$$

$$\leq 4\eta^2 L^2 \frac{1}{n} \sum_{i \in [n]} \left\| \frac{1}{n} \sum_{j \neq i} \nabla f(x_{t-2}, z_{j,t-2}) - \mathbb{E}[\frac{1}{n} \sum_{j \neq i} \nabla f(x_{t-2}, z_{j,t-2})] \right\|^2$$

$$+ 4\eta^2 L^2 \frac{1}{n} \sum_{i \in [n]} \left\| \mathbb{E}[\frac{1}{n} \sum_{j \neq i} \nabla f(x_{t-2}, z_{j,t-3})] - \frac{1}{n} \sum_{j \neq i} \nabla f(x_{t-2}, z_{j,t-3}) \right\|^2$$

$$+ 4\eta^2 L^2 M^2 \|x_{t-2} - x_{t-3}\|^4$$

$$\leq 8\eta^2 L^2 \frac{V_1}{n} + 4\eta^2 L^2 M^2 \|x_{t-2} - x_{t-3}\|^4$$

$$\leq 8\eta^2 L^2 \frac{V_1}{n} + 4\eta^2 L^2 M^2 \|\eta g_{t-2}\|^4$$

$$\leq 8\eta^2 L^2 \frac{V_1}{n} + 4\eta^6 L^2 M^2 V_2^2.$$

For ② , we have
②

$$= 2\eta^2 L^2 \frac{1}{n} \sum_{i \in [n]} \left\| \frac{1}{n} \sum_{j \neq i} \nabla^2 f(x_{t-3}, z_{j,t-3})(x_{t-2} - x_{t-3}) - \lambda \left( \frac{1}{n} \sum_{j \neq i} (g_{j,t-2}) \right) \left( \frac{1}{n} \sum_{j \neq i} (g_{j,t-2}) \right)^\top (x_{t-2} - x_{t-3}) \right\|^2$$

$$\leq 2\eta^2 L^2 (1-\lambda)^2 \frac{1}{n} \sum_{i \in [n]} \left\| \frac{1}{n} \sum_{j \neq i} \nabla^2 f(x_{t-3}, z_{j,t-3})(x_{t-2} - x_{t-3}) \right\|^2$$

$$+ 2\eta^2 L^2 \lambda^2 \frac{1}{n} \sum_{i \in [n]} \left\| \frac{1}{n} \sum_{j \neq i} \nabla^2 f(x_{t-3}, z_{j,t-3})(x_{t-2} - x_{t-3}) - \left( \frac{1}{n} \sum_{j \neq i} (g_{j,t-2}) \right) \left( \frac{1}{n} \sum_{j \neq i} (g_{j,t-2}) \right)^\top (x_{t-2} - x_{t-3}) \right\|^2$$

$$\leq 2\eta^2 L^2 (1-\lambda)^2 L^2 \|x_{t-2} - x_{t-3}\|^2 + 2\eta^2 L^2 \lambda^2 \Delta^2 \|x_{t-2} - x_{t-3}\|^2$$

$$\leq (2\eta^2 L^4 (1-\lambda)^2 + 2\eta^2 L^2 \lambda^2 \Delta^2) \|\eta g_{t-2}\|^2$$

$$\leq 2\eta^4 L^2 V_2 (L^2 (1-\lambda)^2 + \lambda^2 \Delta^2).$$

Putting all the ingredients together, we have

$$\mathbb{E}[F(x_t)]$$

$$\leq F(x_{t-1}) - \frac{\eta}{2} \|\nabla F(x_{t-1})\|^2$$

$$+ \frac{\eta}{2} \|\nabla F(x_{t-1}) - \mathbb{E}[g_t]\|^2 + \frac{L\eta^2}{2} \mathbb{E}\left[ \|g_t - \mathbb{E}[g_t]\|^2 \right]$$

$$\leq F(x_{t-1}) - \frac{\eta}{2} \|\nabla F(x_{t-1})\|^2$$

$$+ \frac{\kappa\eta}{2} [8\eta^2 L^2 \frac{V_1}{n} + 4\eta^6 L^2 M^2 V_2^2 + 2\eta^4 L^2 V_2 (L^2 (1-\lambda)^2 + \lambda^2 \Delta^2)] + \frac{L\eta^2}{2} \frac{V_1}{n}.$$

By re-arranging the terms, we have

$$\|\nabla F(x_{t-1})\|^2$$

$$\leq \frac{2\mathbb{E}[F(x_{t-1}) - F(x_t)]}{\eta}$$

$$+ \kappa (8\eta^2 L^2 \frac{V_1}{n} + 4\eta^6 L^2 M^2 V_2^2 + 2\eta^4 L^2 V_2 (L^2 (1-\lambda)^2 + \lambda^2 \Delta^2)) + \frac{L\eta V_1}{n}.$$

By telescoping and taking total expectation, after $T$ iterations, we have

$$\frac{1}{T} \sum_{t=1}^T \mathbb{E}\left[ \|\nabla F(x_{t-1})\|^2 \right]$$

$$\leq \frac{2\mathbb{E}[F(x_0) - F(x_*)]}{\eta T}$$

$$+ \kappa (8\eta^2 L^2 \frac{V_1}{n} + 4\eta^6 L^2 M^2 V_2^2 + 2\eta^4 L^2 V_2 (L^2 (1-\lambda)^2 + \lambda^2 \Delta^2)) + \frac{L\eta V_1}{n}.$$

$\square$

# B  Proof of Theorem 1

**Theorem 1.** *In training pipeline with no stall, if some forward layer is stale, then all its preceding forward layers are stale.*

*Proof.* We prove this by contradiction. Given staleness for $i$-th forward layer, it satisfies the layer stale condition:

$$c(u_i) < c(v_i) + v_i. \tag{2}$$

Suppose there exists $k$-th layer which is not stale, where $k < i$. According to comm-forward dependency condition, we have

$$c(u_k) \geq c(v_k) + v_k. \tag{3}$$

Since $k < i$, according to forward dependency, we have

$$c(u_k) < c(u_i). \tag{4}$$

Under no preemption assumption, $c(v_i) + v_i \leq c(v_k)$. Combine equations 4 3, we have

$$c(u_i) > c(u_k) \geq c(v_k) + v_k \geq c(v_i) + v_i + v_k > c(v_i) + v_i,$$

which is in contradiction with equation 2. $\square$

# C  Optimized Runtime

## C.1  Weight Update

**Overhead of many small kernel launch.** The default implementation of the weight update function in the ML framework backend involves multiple CUDA kernel launch overhead for each independent parameter. The weight update occurs upon the arrival of each synchronized gradient, which leads to many fragments of small kernels for each parameter. Each kernel launch, regardless of the real computation complexity, incurs some fixed overhead, such as kernel latency, kernel overhead, CPU launch overhead and additional overhead [32]. In general, these kernels cannot be fused together considering different completion time of gradients, while setting a barrier for the completion of all gradient synchronization incurs large waiting overhead. This issue is enlarged with staleness mitigation methods, which introduce extra weight updates and multiple staleness data movement.

**Our Solution: Kernel Fusion.** However, for the staleness pipeline, the weight update kernels could be fused since the input tensors are stored in the staleness buffer in the last step. After backward propagation, the communication operations are executed to synchronize the gradients. The results of synchronized stale gradients are stored in the staleness buffer, while others are directly applied to update the weights on arrival. And the results of stale gradients in the last step are retrieved and the corresponding weight updates kernels are fused together to reduce the overhead.

## C.2  Double Buffer Optimization

**Extra buffer overhead.** There are two staleness buffers for each parameter in SAPipe to enable the cross-iteration execution: the read buffer for computation pipeline reading gradients from previous step and the write buffer for communication pipeline writing synchronized gradients in current step. The double staleness buffer setup increases the GPU memory consumption and incur memory copy overhead to update the reading buffer in each iteration.

**Our Solution: Double Buffer System and Buffer Sharing.** We adopt *double buffer system* technique to omit the memory copy overhead between double staleness buffers. The reading and writing of the staleness buffers shuffles at each training step. The communication service alternate different buffers to write the synchronization results, while the forward computation can read stored data from the available buffer at each step without colliding. This omits the memory copy overhead between two buffers used in different pipeline stages.

To further minimize the memory consumption of staleness buffer, we explore to share the read and write buffer between communication and computation stages. When the read buffer finishes passing the previous step's gradient to the computation backend [6], the synchronized gradient in current step

---

[6]The buffer reading time here denotes the time interval between backward and forward propagation, which consists of dependency awaiting and data transferring time.

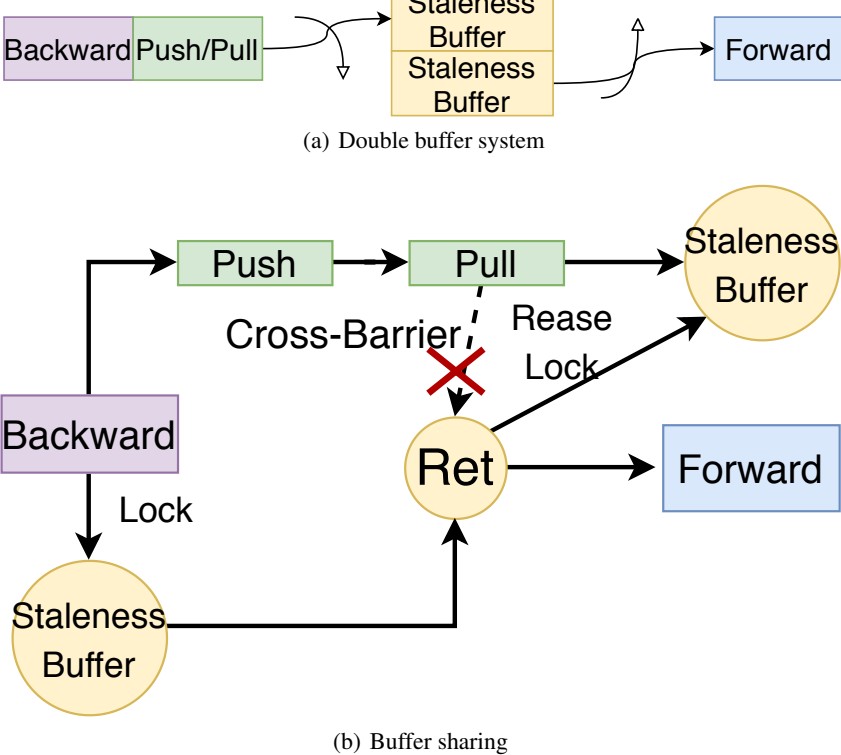

(a) Double buffer system

(b) Buffer sharing

Figure 6: Optimize staleness buffer.

could be allowed to update the reading buffer without being stored in extra memory (see Figure 6 (b)). A dependency is added to avoid race condition that the buffer could be updated only after it has been read in current step. However, this dependency may cause waiting time in the communication pipeline, harming the throughput potentially. Hence, we share the staleness buffer only for the communication-intensive tasks, where the communication time is larger than the buffer reading time.

### C.3 Buffer Switch

**Transferring overhead of staleness buffer.** The staleness buffers are involved in both communication and computation pipelines: 1) the results of synchronized gradient are written into the staleness buffer; 2) the computation stage reads gradients from the staleness buffer for parameter updates. Since the NIC is connected to the PCIe of one of the CPU socket and the computation is on the GPU device, there exists transferring overhead between host and device memory even with only one staleness buffer. The buffer transferring overhead between CPU and GPU is not negligible, considering the slow PCIe bandwidth and large number of parameters.

**Our Solution: Buffer Switch.** However, the buffer transferring overhead between CPU and GPU can be hidden behind pipeline stages of SAPipe. The communication and computation pipelines pipeline of SAPipe might not be balanced due to different communication-to-computation ratios for DNN training jobs. Hence, the staleness buffer could be placed close to the faster pipeline stage for balanced training pipeline. To reduce the transferring overhead, we dynamically switch the location of the staleness buffer according to the duration of communication and computation pipeline stages. Given a DNN training job, SAPipe will profile the completion time of communication ($v_i$) and computation ($b_i$ and $u_i$) through warm-up steps, as well as the buffer transferring overhead $\delta$, and determine the location of the staleness buffer for all gradients: GPU buffer for low communication-to-computation ratio (*e.g.*, $\frac{\sum_i v_i + \delta}{\sum_i (b_i + u_i)} \leq 1$), and CPU buffer for the opposite case (*e.g.*, $\frac{\sum_i v_i}{\sum_i (b_i + u_i) + \delta} > 1$).

The intuition behind is to remove the transferring overhead from slower pipeline stage so as to balance the training pipeline.

**Algorithm 4** Determine the start time of comm ops.

---

**Input:** Forward DAG dependency graph $G = (B, V, E)$, where $b_n$ is the first backward layer;
**Output:** Start time $c(v_i)$ of communication operators.

1:   $S \leftarrow \{b_m\}$
2:   $t \leftarrow c(b_m)$
3:   $r(c_m) \leftarrow 0$
4:   **while** $S$ is not empty **do**
5:      $b_i \leftarrow \text{RANDOMSELECT}(S)$
6:      $S \leftarrow S - \{b_i\}, S \leftarrow S + \{b_j | (b_i, b_j) \in E\}$
7:      $r(v_i) \leftarrow t$
8:      $t \leftarrow t + b_i$
9:   $R \leftarrow \text{SORT}(r(c_1), \ldots, r(v_m))$ // in descending order
10: $t \leftarrow 0$
11: **while** $R$ is not empty **do**
12:      $r(v_k) \leftarrow R[-1]$
13:      $S \leftarrow R.where(r(v_i) <= t) + v_k$
14:      $i = \text{GETMININDEX}(S)$
15:      $S \leftarrow S - v_i$
16:      $c(v_i) \leftarrow t$
17:      $t \leftarrow t + v_i$
18:      $R \leftarrow R - r(v_i)$

---

## D    Partial Staleness on DAG Model

In a more general case where the forward and backward propagation cannot be linearized as a chain of computation across layers, Theorem 1 does not hold and the searching algorithm for sequential models falls short in finding optimal solution. Solving the optimal stale gradient problem for general DAG models is NP-hard. Hence, we present the prototypical heuristic for finding the partial stale gradients for DAG model, as shown in Algorithm 5.

Algorithm 4 determines the execution order and start time for each communication operation, and Algorithm 5 finds the optimal solution of stale gradients. Line 1-8 of Algorithm 4 output the time of each communication operation that is ready to be launched, which is the completion time of their corresponding backward operations. By default, the execute order of available backward operations is decided randomly as in Line 5. Line 9 sorts the ready time of communication operations in descending order, and Line 5-18 determines the launching time for all communication operations. Line 13 selects the candidate communication operations that are available at current scheduling time unit, and Line 14 determines the one with minimal index in ready time set to be launched.

Algorithm 5 further determines the staleness of all gradients according to the scheduled communication operations and forward computation. Line 4 randomly selects one available forward operation $u_i$ from set $S$, and Line 5 updates set $S$ with the neighbors of $u_i$. Line 6-10 checks the stale condition of selected forward operation $u_i$: if its dependent communication operation $v_i$ has not been finished at the scheduling time, then $v_i$ is stale; otherwise, if $v_i$ is finished, $u_i$ can be computed immediately and $v_i$ is non-stale.

## E    Discussion

SAPipe has some requirements for superior model performance:

• Our algorithm for selecting partial staleness can only be used in sequential models, though the staleness compensation and runtime optimizations can be directly applied to a more complicated DAG model. Extending SAPipe to complicated DAG models is our future work.

• The throughput improvement depends on the communication-to-computation ratios. When training models with relatively higher (but no greater than 1) communication-to-computation ratios, SAPipe can achieve higher speedups than the baseline due to higher overlapping potentials.

**Algorithm 5** Searching partial staleness for DAG.

---

**Input:** Forward DAG dependency graph $G = (U, E)$, where $u_1$ is the first forward layer; starting time for communication operators $c(v_i), \forall i = 1, \ldots, m$; starting time for the first forward layer $c(u_1)$

**Output:** Partial staleness for each layer, $x_i, \forall i = 1, \ldots, m$.

1:   $S \leftarrow \{u_1\}$
2:   $t \leftarrow c(u_1)$
3: **while** $S$ is not empty **do**
4:     $u_i \leftarrow \text{RANDOMSELECT}(S)$
5:     $S \leftarrow S - \{u_i\}, S \leftarrow S + \{u_j | (i, j) \in E\}$
6:     $t \leftarrow t + u_i$
7:     **if** $t < c(v_i)$ **then**
8:       $x_i \leftarrow 1$
9:     **else**
10:      $x_i \leftarrow 0$

---

• Theoretically, many factors can affect accuracy, which depends on the properties of the datasets and model structures. Our methods can achieve higher performance under certain conditions, e.g., low gradient variance, low gradient diversity, and good smoothness (as discussed in Remark 7 of Section 4.2).

• The performance of staleness mitigation methods varies in different models and datasets. The best choice of the mitigation methods depends on the choice of hyperparameters and some unknown constant values such as smoothness, gradient diversity and variance. This is the limitation and future work of our paper.

## F   Experiments

### F.1   Datasets Details

We train CV models on two datasets: (1) CIFAR-10 dataset [16], which consists of 50,000 training images and 10,000 test images with 10 classes, and (2) ImageNet dataset [17], which contains 1,281,167 training and 50,000 validation images with 1000 classes. We fine-tunes the pretrained GPT-2 model on (3) WikiText-2 language modeling dataset [20], which is a collection of over 2 million tokens extracted from the set of articles on WikiPedia. Transformer model is trained on (4) Multi30K dataset [8], which is a English-to-German multimodel translation dataset with 29,000 training and 1,000 test sentences.

### F.2   Magnified Figures and Extra Experiments

As shown in Figure 9, in most cases, without the proposed staleness mitigation methods, the vanilla PipeSGD has significant regression in accuracy/perplexity compared to the non-stale baseline, when the same number of iterations are executed. Additionally, we see that PipeSGD has much worse converged results than the baseline in NLP models. This may result from the more complicated models, which are more sensitive to the staleness. Overall, SAPipe has comparable converge rate and final converged results across four models.

Figure 10(a) and (b) show the throughput speedup and staleness mitigation with partial staleness. With partial staleness, SAPipe further reduces the negative impact of staleness on model performance, and improves the accuracy by 1.28% and 2.4% for VGG16 and ResNet50, respectively. We observe divergence when training PipeSGD with ResNet50 on CIFAR-10 dataset, while our staleness mitigation methods greatly solves this issue.

### F.3   Divergence on PipeSGD

We use 2 GPUs to train VGG16 and ResNet50 on CIFAR-10 dataset, and observe severe convergence problems of PipeSGD, as shown in Figure 11. PipeSGD diverges on ResNet50 model with 1-step staleness, while SAPipe mitigates the staleness problem and converges as fast as the BytePS baseline.

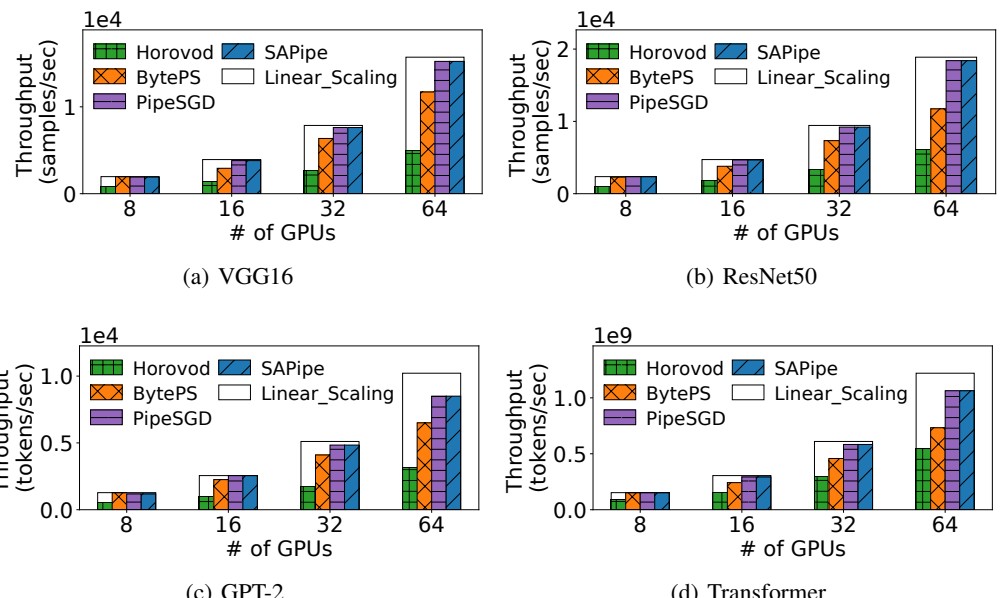

Figure 7: Training throughput.

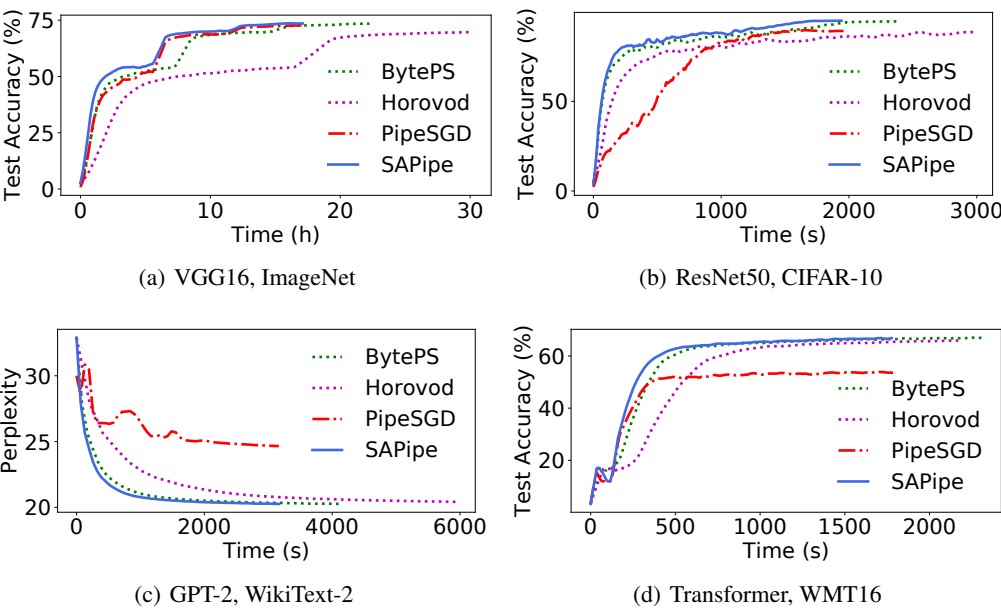

Figure 8: Convergence of different models. The x-axis is wall-clock training time, and the y-axis is perplexity (lower is better) for GPT-2, and test accuracy (higher is better) for others.

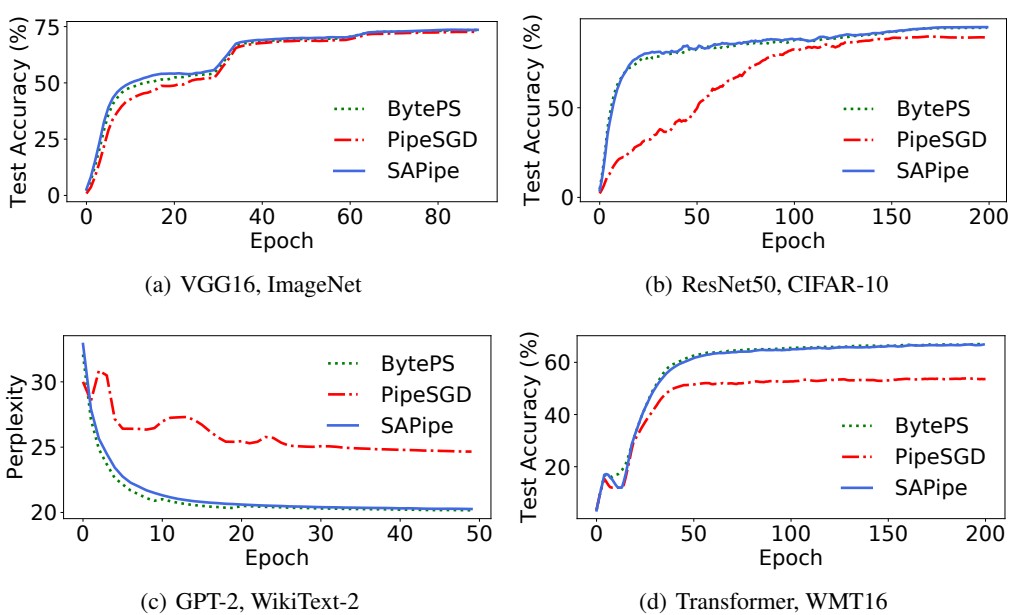

(a) VGG16, ImageNet

(b) ResNet50, CIFAR-10

(c) GPT-2, WikiText-2

(d) Transformer, WMT16

Figure 9: Convergence of different models. The x-axis is the number of epochs.

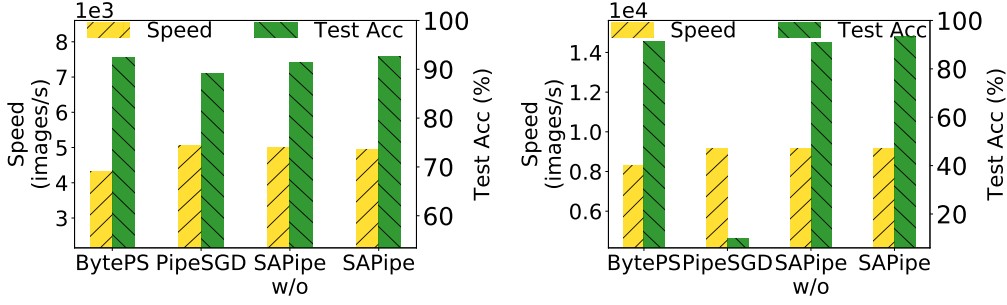

(a) VGG16 with partial staleness. "SAPipe w/o" de-
notes SAPipe without partial staleness.

(b) ResNet50 with partial staleness. "SAPipe w/o"
denotes SAPipe without partial staleness.

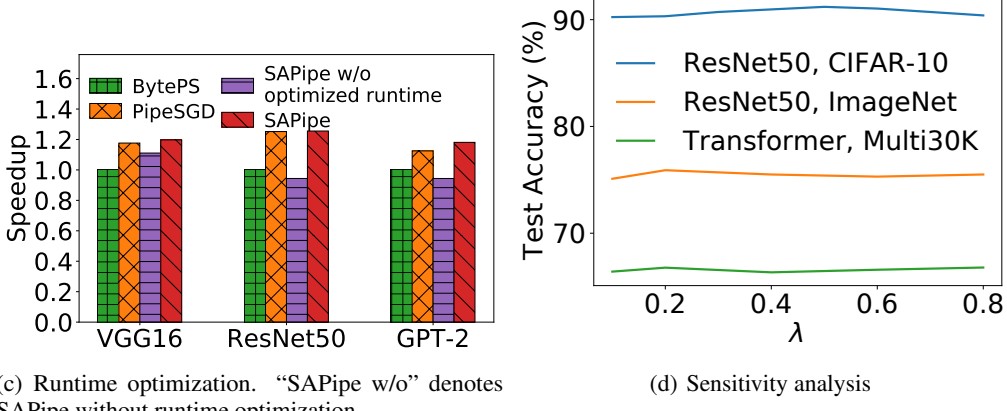

(c) Runtime optimization. "SAPipe w/o" denotes
SAPipe without runtime optimization.

(d) Sensitivity analysis

Figure 10: Deep dive in SAPipe.

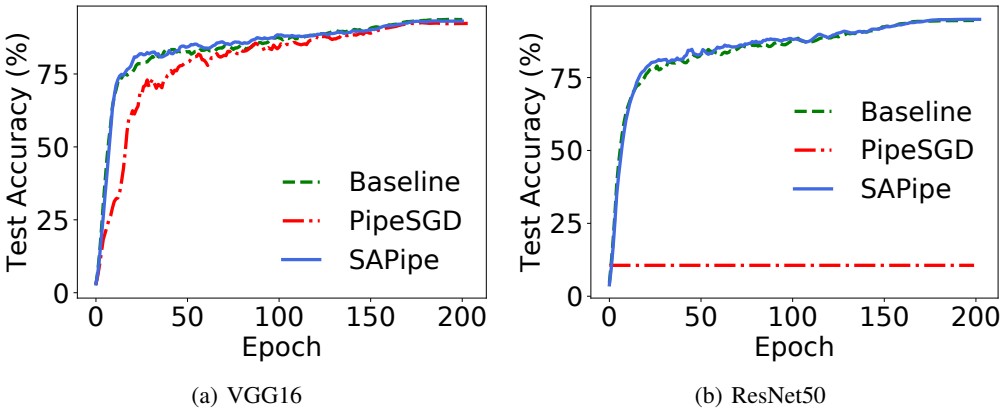

(a) VGG16       (b) ResNet50

Figure 11: Training on CIFAR-10 with 2 GPUs. X-axis is the training epoch, and y-axis is the test accuracy.