# OpenReview forum: "SAPipe: Staleness-Aware Pipeline for Data Parallel DNN Training"
_NeurIPS.cc/2022/Conference — NeurIPS 2022 Accept_

### Official Review · Reviewer_5UGd · 2022-07-11

**Rating:** 6
**Confidence:** 4
**Soundness:** 2 fair
**Presentation:** 3 good
**Contribution:** 2 fair

**Summary:**

Synchronous data parallelism is widely adopted, but can suffer from poor scaling due to excessive communication overheads. This paper proposes using stale weight updates (weight gradients computed using the not latest weights) to better overlap computation and communication. It discusses various mitigation techniques to get bounded-stale data parallelism to work well in practice.

**Questions:**

- Are the ideas of delay compensation and weight prediction new? Don't these papers propose similar ideas for pipeline parallelism without flushes (which shouldn't make a difference): PipeMare [https://proceedings.mlsys.org/paper/2021/file/6c8349cc7260ae62e3b1396831a8398f-Paper.pdf] and Kosson et al. [https://arxiv.org/pdf/2003.11666.pdf].
- Why does the optimization problem at the top of page 4 have a runtime complexity of O(m^2)?
- What batch size was used? Communication overhead is smaller with larger batch sizes.
- What is the effect of hardware (compute accelerator and network) on performance?
- What is the effect of model on performance?
- How much worse convergence rate (accuracy vs. number of iterations) does SAPipe have compared to synchronous DP (BytePS or Horovod)?
- How much do the proposed mitigation techniques (delay compensation and weight prediction) help with convergence? What if I don't use any mitigation techniques? How badly does this do?

**Limitations:**

The authors did not discuss limitations of their work. Some discussion of situations where they expect SAPipe to not be a useful solution (e.g., particular types of models, hardware deployments, optimizers, or other situations where their theoretical analysis breaks down).

**Strengths And Weaknesses:**

#### Strengths
- The paper offers an alternative to synchronous data parallelism in situations where the network is not fast enough to hide the cost of communication.
- The paper is well written and clear.
- Distributed training performance is an important problem in Machine Learning Systems, and this paper proposes one more possible solution to the problem.

#### Weaknesses
- Evaluation is not entirely convincing: for example, Figure 4 shows accuracy vs. time, but I would have also liked to see accuracy vs. iteration to see the impact on convergence speed.
- Baselines seem if-fy: why does ResNet-50 scale worse than GPT-2? I expect the convolutional layers in a ResNet-50 model to be much more amenable to data-parallel-style communication compared to a GPT-2 model with a lot of linear layers in the attention layers. VGG-16 doesn't scale well exactly for these reasons.

---

> ### Author Response · Authors · 2022-08-02
> **Major Response to Reviewer 5UGd**
>
> 1. why does ResNet-50 scale worse than GPT-2?
>
>    **A**: ResNet-50 and GPT-2 have similar poor scaling ratios (real throughput with 64 GPUs over throughput with 8 GPUs times 8) in our experiment, 0.64 and 0.63, respectively (see Figure 3(b) and 3(c)). Though ResNet-50 involves less communication compared to GPT-2,  the inter-server communication still becomes the bottleneck when scaling to over 32 workers.
>
> 2. Are the ideas of delay compensation and weight prediction new? Related work: PipeMare and Kosson et al.
>
>    **A**: The algorithms proposed in this paper are novel in several aspects as follows:
>
>       - This paper focuses on data parallelism, while PipeMare/Kosson et al. uses model parallelism. Applying staleness mitigation to data parallelism is novel and different from previous work.
>
>       - Weight prediction (WP) is a general concept to mitigate weight inconsistency, and this paper provides several WP methods to solve the staleness issue on data parallelism. SAPipe-WP-OPT2 is similar to linear weight prediction used in previous work, while SAPipe-WP-OPT1 and SAPipe-WP-OPT3 are absent in PipeMare/Kosson et al., and achieve better results in our experiments.
>
>       - The theoretical analysis of convergence is one of our main contributions, while there is no convergence analysis in PipeMare/Kosson et al.. Furthermore, it is non-trivial to prove the convergence of SAPipe with various WP.
>
>       - Additionally, we modify the delay compensation (DC) method using the full-matrix form to avoid the additional error caused by the diagonal approximation, compared to the original paper [33], as we explained in Section 3.2, Line 130-140. Furthermore, the corresponding theoretical analysis of convergence is also different from the original paper.
>
> 3. Why does the optimization problem at the top of page 4 have a runtime complexity of O(m^2)?
>
>    **A**: For each k, we compute summation for O(m) times in constraints. And the maximal value of k is m.
>
> 4. What batch size was used?
>
>    **A**: We specify the batch size for each model in Section 5.1 (Line 262).
>
> 5. What is the effect of hardware  on performance?
>
>    **A**: SAPipe is suitable for hardware deployment with roughly comparable computation and communication time. We only have one type of hardware, Tesla V100, in our lab. Studying the effect of different hardware could be our future work.
>
> 6. What is the effect of model on performance?
>
>    **A**: The properties of models, e.g., the model structure and the number of parameters, affect the constant values in our assumptions, such as smoothness ($L$ in Assumption 1), gradient variance ($V_1$ in Assumption 2), gradient diversity ($\rho$ in Assumption 3), etc. However, studying the exact effect of the model properties on our assumptions is beyond the scope of this paper, which could be our future work.
>
> 7.  How much worse convergence rate does SAPipe have compared to synchronous DP?
>
>     **A**: We have added the figures with convergence rate comparison (accuracy vs. #epoch) in Figure 9 in appendix.  As shown in Figure 9, in most cases, without the proposed staleness mitigation methods, the vanilla PipeSGD has significant regression in accuracy/perplexity compared to the non-stale baseline, when the same number of iterations are executed. When the staleness mitigation is used, SAPipe has the same convergence rate compared to synchronous DP ("SAPipe" in Figure 9 is the one with the best choice of staleness mitigation method).
>
> 8. How much do the proposed mitigation techniques help with convergence?
>
>    **A**: PipeSGD is the 1-stale pipeline method without any mitigation techniques, causing obvious impact to convergence as shown in Table 2, Figure 4, and Figure 9 in appendix.
>
> 9. The authors did not discuss limitations of their work.
>
>    **A**: One limitation of our work is lack of finding the best staleness mitigation options for different models and datasets. We previously put a brief discussion of this limitation in Remark 7. Note that Remark 3\~7 all have mentioned that the proposed staleness mitigation methods (DC/WP-OPT1/OPT2/OPT3) has lower error compared to vanilla PipeSGD conditional on certain constant values depends on the models and data. For example, Line 224-225 states that better convergence of SAPipe-WP-OPT1 requires small gradient divergence, otherwise the vanilla PipeSGD would be better.  In the revised version, we have also added such discussion in Line 289-295 and 335-337.
>
>    In section 6 related work, we also mention that we haven't tried to combine pipelined training with gradient compression. These 2 methods are orthogonal and could be easily combined, which is another limitation of this work.
>
>    For optimizers, since our proposed staleness mitigation methods are applied directly to the gradients before entering the optimizer, they are compatible to most of the popular 1st-order methods, such as SGD and Adam, but not compatible to the 2nd-order methods.

---

> > ### Comment · Reviewer_5UGd · 2022-08-08
> > **Response to comments**
> >
> > Thank you for the detailed responses to my questions. Some comments / follow-up questions below.
> > - Thank you for pointing out the differences compared to staleness mitigation techniques used in pipeline parallelism. The differences make sense to me.
> > - I did not see the same results when trying to scale GPT-2 and ResNet-50 in the past on comparable hardware on AWS. What was the setup? To confirm, the total batch size is reported per-GPU batch size (e.g., 128 images for ResNet-50) times the number of GPUs used?
> > - When I asked for the effect of hardware on performance, I was looking for qualitative estimates rather than new experiments.
> > - I was asking more about the effect of models on throughput. For example, will your technique work better for VGG-16 or ResNet-50, which have different computation-communication profiles when scaling using data parallelism?
> > - For the scaling experiments, doesn't it make more sense to fix the global batch size and then change the per-GPU microbatch size and degree of gradient accumulation accordingly, since the batch size affects semantics. Having said that, 80 tokens per GPU seems very small.
> > - Thank you for the pointers to the comparisons to SAPipe without staleness mitigation (PipeSGD).
> > - Can I use SAPipe for any model and expect superior time-to-accuracy compared to synchronous methods?
> > - Why is Horovod slightly different from BytePS on the accuracy vs. epochs graph?
> > - As reviewer 9R6e points out, I would also like to see comparisons to PyTorch's DDP.

---

> > > ### Author Response · Authors · 2022-08-09
> > > **Response to Reviewer 5UGd's 2nd Response**
> > >
> > > 1. I did not see the same results when trying to scale GPT-2 and ResNet-50 in the past on comparable hardware on AWS. What was the setup? To confirm, the total batch size is reported per-GPU batch size (e.g., 128 images for ResNet-50) times the number of GPUs used?
> > >
> > >    **A**: Our setup can be found at Section 5.1 (Line 258). A lot of factors can affect the scalability. Even with same network bandwidth, different network topology can still result in different scalability. We run our experiments on our own clusters, so the results could be different from those on AWS.
> > >
> > >       Yes, the total batch size is reported per-GPU batch size times the number of GPUs used.
> > >
> > > 2. When I asked for the effect of hardware on performance, I was looking for qualitative estimates rather than new experiments. I was asking more about the effect of models on throughput. For example, will your technique work better for VGG-16 or ResNet-50, which have different computation-communication profiles when scaling using data parallelism?
> > >
> > >    **A**: The benefit of using PipeSGD/SAPipe mainly depends on the communication-to-computation ratio, which is affected by many factors, such as the computation power, the network bandwidth, and the model sizes. PipeSGD/SAPipe is suitable for hardware deployment with roughly comparable computation and communication time. If the communication time is too long, there will be limited overlapping space to reduce the communication overhead; If the computation time is too long, the communication overhead may be negligible or already covered by the forward and backward pass.
> > >
> > >    SAPipe has better improvement on models with relatively higher (but no greater than 1) communication-to-computation ratio, 57% for VGG16 and 31% for ResNet50, respectively.
> > >
> > > 4. For the scaling experiments, doesn't it make more sense to fix the global batch size and then change the per-GPU microbatch size and degree of gradient accumulation accordingly, since the batch size affects semantics. Having said that, 80 tokens per GPU seems very small.
> > >
> > >     **A**:  Our goal is to speedup the training throughput with more computation resources, while fixing the global batch size mainly benefits the setups with a small number of GPUs. We follow the previous related work to setup the scaling experiments (fixing per-GPU micro-batch size), e.g., ZeRO-Offload [https://www.usenix.org/conference/atc21/presentation/ren-jie] and BytePS [https://www.usenix.org/system/files/osdi20-jiang.pdf].
> > >
> > >     For GPT-2, we use the default hyper-parameters in the examples of BytePS [https://github.com/byteps/examples]. GPT-2 is a large model, and there occurs out-of-memory issues for GPU memory when we attempt to increase the batch-size.
> > >
> > > 5. Can I use SAPipe for any model and expect superior time-to-accuracy compared to synchronous methods?
> > >
> > >    **A**: No, SAPipe has some requirements for superior performance:
> > >
> > >       a) Our algorithm for selecting partial staleness can only be used in sequential models, though the staleness compensation and runtime optimizations can be directly applied to a more complicated DAG model.
> > >
> > >       b) The throughput improvement depends on the communication-to-computation ratios. When training models with relatively higher (but no greater than 1) communication-to-computation ratios, SAPipe can achieve higher speedups than the baseline due to higher overlapping potentials.
> > >
> > >       c) Theoretically, many factors can affect accuracy, which depends on the properties of the datasets and model structures. Our methods can achieve higher performance under certain conditions, e.g., low gradient variance, low gradient diversity, and good smoothness. We discuss the details in remark 7 of Section 4.2 (Line 248).
> > >
> > > 6. Why is Horovod slightly different from BytePS on the accuracy vs. epochs graph?
> > >
> > >    **A**: Horovod does not appear in accuracy vs. epochs graph (Figure 9 in appendix), since both Horovod and BytePS do not modify the training algorithm. Could you please inform us which figure you are referring to?
> > >
> > > 7. As reviewer 9R6e points out, I would also like to see comparisons to PyTorch's DDP.
> > >
> > >    **A**:  In our implementation, we are using the updated version of BytePS, which includes similar optimizations of state-of-the-art version of PyTorch. In some of our earlier investigations, we found that BytePS is still better than state-of-the-art version of PyTorch. For example (throughput of training wav2vec model, batch-size 4500):
> > >
> > >     | #GPU        | 1     | 8     | 16     | 32     |
> > >     |-------------|-------|-------|--------|--------|
> > >     | PyTorch-DDP | 12429 | 90059 | 123387 | 245256 |
> > >     | BytePS      | 12468 | 91520 | 143717 | 311123 |
> > >
> > >    Hence, we use BytePS as our baseline.

---

> > > > ### Comment · Reviewer_5UGd · 2022-08-09
> > > > **Thanks for the responses**
> > > >
> > > > Thanks for the responses. I will update my score.
> > > > I read Figure 9 wrong. However, why does SAPipe perform (marginally) better than fully synchronous for some tasks (VGG-16 and ResNet-50)? Also, this is a nit, but please add more yticks when necessary (e.g., ResNet-50 in Figure 9).
> > > > I think adding some discussion to the final version of the paper would be good (especially the part about when SAPipe does well and when it doesn't). The additional experiments are appreciated.

---

> > > > > ### Author Response · Authors · 2022-08-10
> > > > > **Thanks for your response**
> > > > >
> > > > > Thanks for your response and suggestion.
> > > > >
> > > > > 1. why does SAPipe perform (marginally) better than fully synchronous for some tasks (VGG-16 and ResNet-50)?
> > > > >
> > > > >    **A**: We do observe that in a few cases, SAPipe performs slightly better than fully synchronous SGD. For example, in Table 2, SAPipe-WP-OPT1 has a little bit better accuracy than BytePS when training VGG-16 on CIFAR-10. We also observe that this phenomenon happens when we use the local gradients in weight prediction (SAPipe-WP-OPT1 and SAPipe-WP-OPT3). Our intuition is that, by using the local gradients, SAPipe-WP may share some advantages with local SGD, which also uses local gradients. There are some recent research indicating that local SGD can perform better than mini-batch SGD under certain conditions (please refer to https://arxiv.org/pdf/1808.07217.pdf and http://proceedings.mlr.press/v119/woodworth20a/woodworth20a.pdf for more details). However, further research with theoretical analysis and empirical evaluation will be needed in our future work to fully understand it.
> > > > >
> > > > > 2. Please add more yticks when necessary (e.g., ResNet-50 in Figure 9). I think adding some discussion to the final version of the paper would be good (especially the part about when SAPipe does well and when it doesn't). The additional experiments are appreciated.
> > > > >
> > > > >    **A**: Thanks for the suggestion. Since the rebuttal revision is closed, we will make revision in the final version as suggested by the reviewer.

---

> ### Author Response · Authors · 2022-08-02
> **Detailed Response to Question 1**
>
> **Q**: Are the ideas of delay compensation and weight prediction new? Don't these papers propose similar ideas for pipeline parallelism without flushes (which shouldn't make a difference): PipeMare [https://proceedings.mlsys.org/paper/2021/file/6c8349cc7260ae62e3b1396831a8398f-Paper.pdf] and Kosson et al. [https://arxiv.org/pdf/2003.11666.pdf].
>
>    **A**: The algorithms proposed in this paper are novel in several aspects as follows:
>    1. This paper focuses on data parallelism, while PipeMare/Kosson et al. uses model parallellism. Applying staleness mitigation to data parallelism is novel and different from previous work. Note that the distributed training mechanism used in model parallelism papers (PipeMare/Kosson et al.) and data parallelism papers (PipeSGD/SAPipe) are totally different. In brief, we summarize the main differences as follows: For PipeMare/Kosson et al: 1) model parallelism; 2) in the same batch, different micro/mini-batches use different versions of model parameters with different staleness for forward-backward; 3) for the same micro/mini-batch, the model parameters used in forward pass and backward pass are different due to the different staleness; 4) weight prediction is used to close the gap between different versions of model parameters in forward pass and backward pass; 5) no theoretical results for convergence. While for SAPipe: 1) data parallelism; 2) in the same batch, different mini-batches or workers use the same version of model parameters, with exactly the same fixed staleness of 1; 3) same model parameters are used in forward pass and backward pass; 4) weight prediction/delay compensation is used to mitigate the fixed staleness of 1; 5) theoretical results for convergence are provided (and note that our convergence proof could not be applied to PipeMare/Kosson et al., due to the difference between data parallelism and model parallelism). In a nutshell, SAPipe has its own unique properties, which results in novel algorithms using delay compensation and weight prediction, and new challenges in the convergence proof. Furthermore, some options such as weight prediction with local gradients (SAPipe-WP-OPT1/OPT3) are infeasible for PipeMare/Kosson et al.
>
>    2. Weight prediction (WP) is a general concept to mitigate weight inconsistency, and this paper provides several WP methods to solve the staleness issue on data parallelism. SAPipe-WP-OPT2 is similar to linear weight prediction used in previous work, while SAPipe-WP-OPT1 and SAPipe-WP-OPT3 are absent in PipeMare/Kosson et al., and achieve better results in our experiments.
>
>    3. The theoretical analysis of convergence is one of our main contributions, while there is no convergence analysis in PipeMare/Kosson et al., regardless of the difference between data parallelism and model parallelism. Furthermore, it is non-trivial to prove the convergence when adding different options of weight prediction to SAPipe. For example, in SAPipe-WP-OPT2, the use of weight prediction with latest synchronized gradient causes a recursive error term (Line 524-529 in Appendix). Such a recursive error term doesn't exist in the previous work of PipeSGD or asynchronous SGD, which is the key to show that SAPipe-WP has lower error bound compared to vanilla PipeSGD/SAPipe without staleness mitigation under certain conditions, as we have discussed in Remark 3, 4, 5, 6 in Section 4.2.
>
>    4. Additionally, we modify the delay compensation (DC) method using the full-matrix form to avoid the additional error caused by the diagonal approximation, compared to the original paper [33], as we explained in Section 3.2, Line 130-140. Furthermore, the corresponding theoretical analysis of convergence is also different from the original paper. In our proof, we remove an unreasonable and impractical assumption from the original paper of DC, which manually sets a "search region" for the model parameters: $\\| x - x' \\|^2_2 \leq \pi^2$, where $x$ and $x'$ are any 2 versions of model parameters in the training sequence of $\\{ x_t, t \in [T] \\}$. The abandonment of this assumption incurs extra difficulty to our convergence proof. To overcome this challenge, we adopt a new technique, where the error bound is established by solving a sequence of recursive inequalities (Line 498-500 in Appendix). Such a new proof procedure can not be found in the previous work.

---

### Official Review · Reviewer_9R6e · 2022-07-11

**Rating:** 6
**Confidence:** 4
**Soundness:** 3 good
**Presentation:** 3 good
**Contribution:** 2 fair

**Summary:**

This paper presents SAPipe, a system to support efficient data parallelism, where communication is effectively hidden within computation with adaptive staleness and corresponding compensations. Both theoretical analysis and empirical study are conducted to verify the effectiveness of the proposed solution.

**Questions:**

N.A.

**Limitations:**

Perhaps some discussion about the scope of models should be considered; for example, can this approach be used for graph neural network training, where the layers in the model are not linearly stacked?

**Strengths And Weaknesses:**

Strengths:
- The idea of improving PipeSGD with adaptive partial staleness is simple but effective.

- Theoretical analysis is provided to justify the design of the algorithm.

- The experiment section is solid, and the performance boost is significant.

Weaknesses:
- Some writing and illustration can be further polished. For example, Figure 1 is a little confusing; for the default pipeline part, it seems that v3 begins before the end of b3 visually, this sees inaccurate without mentioning any potential other optimization, e.g., communicating in a thinner granularity.

- Some reasonable baseline approaches are missing. For example, PyTorch-DDP should be included because it is a very popular data-parallel implementation and provides efficient system optimizations such as bucketing and communication overlapping.

---

> ### Author Response · Authors · 2022-08-02
> **Response to Reviewer 9R6e**
>
> 1. Figure 1 is a little confusing; for the default pipeline part, it seems that v3 begins before the end of b3 visually, this sees inaccurate without mentioning any potential other optimization, e.g., communicating in a thinner granularity.
>
>     **A**: Thanks for the comment. The arrows denote dependencies between two operators. We have specified this in the caption of Figure 2 to avoid confusion (Line 114).
>
> 2. PyTorch-DDP should be included as a baseline, because it is a very popular data-parallel implementation and provides efficient system optimizations such as bucketing and communication overlapping.
>
>     **A**: We use BytePS as the non-stale baseline, because it has higher training throughput than PyTorch-DDP (referred to as state-of-the-art all-reduce implementation) with more communication optimization techniques, as shown in the paper of BytePS [https://www.usenix.org/system/files/osdi20-jiang.pdf].
>
> 3. Perhaps some discussion about the scope of models should be considered; for example, can this approach be used for graph neural network training, where the layers in the model are not linearly stacked?
>
>     **A**: Our method could be easily extended to GNN, but this may be unnecessary. Since GNN is shallow with much fewer parameters than DNN, its bottleneck is usually at data preprocessing (computational graph sampling and feature retrieving), which has negligible gradient synchronization overhead.

---

> > ### Comment · Reviewer_9R6e · 2022-08-08
> > **Thanks for your response!**
> >
> > Thank the author for their detailed response! Here are some follow-ups:
> >
> > 1. Thanks for the clarification!
> >
> > 2. I think recent version of PyTorch-DDP includes a series of system optimization, including tensor flattening and bucketing, overlapping communication and computation, etc., which was report at VLDB 2020 (https://www.vldb.org/pvldb/vol13/p3005-li.pdf), I am not sure this state-of-the-art version of PyTorch was included in the comparison reported in the BytePS paper. If the author confirmed similar optimizations also exists in BytePS, I would agree that the comparison with BytePS would also support the statement in the empirical study.
> >
> > 3. Sorry for triggering some confusion. To be more clear about the questions, I was trying to ask if the proposed optimization can be adopted to DNN represented by a more complicated DAG (instead of simply linearly stacked blocks, such as VGG, ResNet40, GPT2)? GNN is just one example, Perhaps U-net is a better example for this question. Lastly, I want to emphasize that, in my opinion, clear statement of the applicable scope of a methodology would be considered as a strength instead of a weakness.

---

> > > ### Author Response · Authors · 2022-08-09
> > > **Response to Reviewer 9R6e's 2nd Response**
> > >
> > > 1. I am not sure this state-of-the-art version of PyTorch was included in the comparison reported in the BytePS paper. If the author confirmed similar optimizations also exists in BytePS, I would agree that the comparison with BytePS would also support the statement in the empirical study.
> > >
> > >
> > >    **A**: Yes, the PyTorch version in the original BytePS paper does not include these system optimizations when comparing with BytePS (we took some time to contact the author to confirm that). However, in our implementation, we are using the updated version of BytePS, which includes similar optimizations of state-of-the-art version of PyTorch. In some of our earlier investigations, we found that BytePS is still better than state-of-the-art version of PyTorch. For example (throughput of training wav2vec model, batch-size 4500):
> > >
> > >     | #GPU        | 1     | 8     | 16     | 32     |
> > >     |-------------|-------|-------|--------|--------|
> > >     | PyTorch-DDP | 12429 | 90059 | 123387 | 245256 |
> > >     | BytePS      | 12468 | 91520 | 143717 | 311123 |
> > >
> > >    Hence, we use BytePS as our baseline.
> > >
> > > 2. To be more clear about the questions, I was trying to ask if the proposed optimization can be adopted to DNN represented by a more complicated DAG (instead of simply linearly stacked blocks, such as VGG, ResNet40, GPT2)?
> > >
> > >
> > >    **A**: Yes, we implied that our work focuses on the sequential models (Line 100), and this is indeed our limitation. Our staleness compensation methods and runtime optimizations can be directly applied to a more complicated DAG model. But the method of selecting partial staleness cannot be directly used in DAG models. In the appendix, we discuss how to extend the partial staleness method for complicated DAG models (see Appendix D).

---

### Official Review · Reviewer_oko6 · 2022-07-15

**Rating:** 5
**Confidence:** 3
**Soundness:** 3 good
**Presentation:** 2 fair
**Contribution:** 3 good

**Summary:**

One of the challenges with scaling data parallel training of neural networks is that the natural dependencies between the forward propagation, backward propagation, and gradient updates limits the benefits of parallel execution.  A common solution in the state of the art is to use stale gradients to allow the gradient updates to overlap with the data motion between parallel ranks.  However, the use of stale gradients typically leads to a lower quality of solution in the trained network.  This paper presents several methods for addressing the staleness of gradients, using three techniques: partial staleness for “earlier” layers of the network, delay gradient compensation, and weight prediction.  Using these techniques they demonstrate that they can achieve runtime performance similar to the SOTA using stale gradients, but with a quality that nearly matches a standard SGD training approach.  The paper also provides both a convergence analysis and experimental analysis with both visual and language models demonstrating the impact of their proposed techniques.

**Questions:**

Figure 1 is introduced early in the text, prior to any significant discussion of partial stale gradient updates, yet it is illustrating partially stale gradients.  The figure, text, and caption should be refined to indicate that it is not showing the state of the art, but actually part of the proposed method.

The implementation of Algorithm 1 is strange.  Specifically, line 6 is weird for the base case.  For it to say “same as t > 1” should be clarified that there is really only a conditional for the PipeSGD case.

Figure 2 doesn’t really convey what the authors probably intended for explaining the algorithm and does not contribute too much as is.  For subfigure 2a, the fact that the parts of the staleness-aware system are all in the lower box but with no real relationships between them and the runtime parts of the system is not helpful to the reader.  Furthermore for 2b, it would be better to represent the communication pattern with some sort of parallel timeline.  Again, with the cyclic dependencies of iterative training, the interplay of the communication isn’t clearly conveyed by 2b.

In the equation below line 111 it is not clear what u is as it doesn’t appear in the table.

On line 119 a figure that clearly shows how partial staleness / gradient updates works would improve the discussion.

Replace Figure 5 analysis with a more challenging example for the ablation study, i.e. ResNet on ImageNet or one of the transformers, or both.

**Limitations:**

The biggest issue is that of the proposed techniques, on the use cases shown, one combination does not always win.  As a result it is not clear if these approaches were to be applied to a new problem area, which set of the staleness compensation techniques should be used.  Without some discussion of this, it is not clear how to leverage this work going forward without actually performing something of this analysis.  Furthermore, the section on sensitivity analysis does not address this.

**Strengths And Weaknesses:**

This paper leverages multiple advancements in the community and integrates them into a common composition and framework, demonstrating that in aggregation they can provide an approach that is performant in terms of both speed and quality of the trained network.  Overall, they clearly indicate where the derived ideas come from, how they were used previously and how they are being integrated into a combined algorithm.  The explanation of the algorithm and visualization via diagram could use some improvement.  Please see the next section for specific comments, however, the reader is left with a general understanding of how the algorithm execute at runtime, but some better figures could significantly improve this.

The evaluation of the algorithm is mixed.  Overall it is good that the authors include both visual and language models, however, using VGG16 on CIFAR-10 for the ablative study is not nearly as interesting as if they had used a more modern and interesting model (e.g. ResNet or GPT-2).  Additionally, the later part of the paper is quite rushed with minimal explanation and analysis of the results.  For example Figure 5 has no analysis in the caption and only a bit in the text.  On a similar note, Figures 3-5 are extremely small and clearly not really meant for a reader to actually read.

One key challenge for this work is that none of the proposed optimization schemes performed the best on all benchmarks, and thus is is not clear a priori as to which technique (Opt1 - 3) should be used when.  This can pose a significant challenge to future application if a researcher would have to conduct an exhaustive test of all optimization methods to determine which may produce the best results for a new application.

---

> ### Author Response · Authors · 2022-08-02
> **Response to Reviewer oko6**
>
> 1. Using VGG16 on CIFAR-10 for the ablative study is not nearly as interesting as if they had used a more modern and interesting model (e.g. ResNet or GPT-2).
>
>     **A**: ResNet and GPT-2 are also shown in the ablation study in Figure 5(b). Ablative study for partial stale experiments with ResNet model can be found in Figure 10(b) in appendix.
>
> 2. The later part of the paper is quite rushed with minimal explanation and analysis of the results. For example Figure 5 has no analysis in the caption and only a bit in the text.
>
>     **A**: We have added the detailed descriptions of experiment results in appendix (E.2, Line 641).
>
> 3. None of the proposed optimization schemes performed the best on all benchmarks. This can pose a significant challenge to future application if a researcher would have to conduct an exhaustive test of all optimization methods to determine which may produce the best results for a new application.
>
>     **A**:  Yes, the performance of staleness mitigation methods varies in different models and datasets. In theoretical analysis, we have also explained that the best choice of the mitigation methods depends on the choice of hyperparameters and some unknown constant values such as smoothness, gradient diversity and variance, which depends on the data and model, as we discussed in Remark 7, Line 251-255.  This is the limitation and future work of our paper. In the revised version, we have also added this discussion Line 293-295 and Line 335-337.
>
> 4. Figure 1 is introduced early in the text, prior to any significant discussion of partial stale gradient updates.
>
>     **A**: Thanks for the comment. We have moved Figure 1 (DNN training pipeline) to Section 3 (Line 114), which is now Figure 2.
>
> 5. Line 6 of Algorithm 1 is strange. For it to say “same as t > 1” should be clarified that there is really only a conditional for the PipeSGD case.
>
>     **A**: Yes, this is only conditional for PipeSGD case.  This line is just for comparison of DNN training pipeline and PipeSGD so that the difference between normal distributed training and PipeSGD is clearer to the readers. For implementation, distributed training doesn't need such a conditional.
>
> 6. For subfigure 2a, the fact that the parts of the staleness-aware system are all in the lower box but with no real relationships between them and the runtime parts of the system is not helpful to the reader.
>
>     **A**: Thanks for the comment. We have modified this figure and added relationships between our algorithm design and runtime optimizations, which is now Figure 1(a) in the revised version.
>
> 7. For 2b, it would be better to represent the communication pattern with some sort of parallel timeline. Again, with the cyclic dependencies of iterative training, the interplay of the communication isn’t clearly conveyed by 2b.
>
>     **A**: The text below this subfigure (now Figure 1(b) in the revised version) shows the timeline.
>
> 8. In the equation below line 111 it is not clear what u is as it doesn’t appear in the table.
>
>     **A**: $u$ is the duration of forward operator, which can be found in Table 1 (in the 3rd row, left column).
>
> 9. On line 119 a figure that clearly shows how partial staleness / gradient updates works would improve the discussion.
>
>     **A**: Thanks for the suggestion. We have moved the figure of training pipeline with partial staleness in this section (Line 114, Figure 2).
>
> 10. Replace Figure 5 analysis with a more challenging example for the ablation study, i.e. ResNet on ImageNet or one of the transformers, or both.
>
>     **A**: ResNet and GPT-2 are also shown in the ablation study in Figure 5(b). Ablative study for partial stale experiments with ResNet model can be found in Figure 10(b) in appendix.
>
> 11. Figures 3-5 are extremely small.
>
>     **A**: Magnified figures can be found in appendix (Figure 7-10).

---

### Meta-Review · Area_Chair_kuQA · 2022-08-22

**Recommendation:** Accept
**Confidence:** Less certain

**Metareview:**

This paper proposes a new algorithm to speed up data-parallel distributed training, focused on mitigating staleness-induced issues that arise when limiting communication between nodes.

All reviewers and myself agree this is a worthwhile contribution, which is backed by both convincing empirical and theoretical results. I consider that the potential novelty concerns that were raised in initial reviews have been addressed by the authors. The main remaining concerns are related to the limitations of the proposed method, that comes with some trade-offs and may not apply to all situations. I believe the authors have adequately answered these concerns by being upfront about these limitations during the discussion period, and I encourage them to make sure this is also clear in the final version of the paper. In spite of these limitations, I believe the novelty and significance of this work meet the bar for acceptance at NeurIPS, since speeding up distributed computations is a very relevant and challenging problem in modern deep learning.

**Award:**

No

---

### Decision · Program_Chairs · 2022-09-14

Accept